# Texture Vector-Quantization and Reconstruction Aware Prediction for Generative Super-Resolution

**Qifan Li    Jiale Zou    Jinhua Zhang    Wei Long    Xingyu Zhou    Shuhang Gu**[*]
University of Electronic Science and Technology of China
qifanli.lqf@gmail.com   shuhanggu@gmail.com

## Abstract

Vector-quantized based models have recently demonstrated strong potential for visual prior modeling. However, existing VQ-based methods simply encode visual features with nearest codebook items and train index predictor with code-level supervision. Due to the richness of visual signal, VQ encoding often leads to large quantization error. Furthermore, training predictor with code-level supervision can not take the final reconstruction errors into consideration, result in sub-optimal prior modeling accuracy. In this paper we address the above two issues and propose a **T**exture **V**ector-**Q**uantization and a **R**econstruction **A**ware **P**rediction strategy. The texture vector-quantization strategy leverages the task character of super-resolution and only introduce codebook to model the prior of missing textures. While the reconstruction aware prediction strategy makes use of the straight-through estimator to directly train index predictor with image-level supervision. Our proposed generative SR model (TVQ&RAP) is able to deliver photo-realistic SR results with small computational cost.

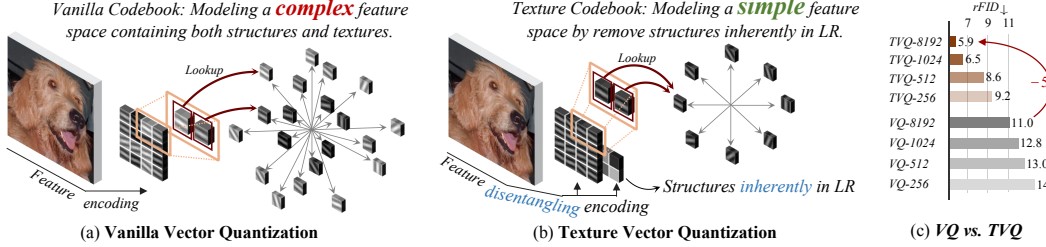

(a) **Vanilla Vector Quantization**    (b) **Texture Vector Quantization**    (c) *VQ vs. TVQ*

Figure 1: **Vanilla VQ *vs*. Texture VQ.** Vanilla VQ directly encode the entire visual feature space, a large codebook is required to capture complex combinations of structure and texture information. Our Texture VQ focuses on modling textures absent in LR inputs, thereby mitigating the difficulty of visual encoding for generative super-resolution. Notably, TVQ achieves significantly better reconstruction performance than the vanilla method across a range of codebook sizes. Experimental details can be found in Section 4.3.

# 1 Introduction

Image super-resolution (SR) aims to reconstruct high-resolution (HR) images from their low-resolution (LR) counterparts. Classical SR methods target at minimizing the Root Mean Square Error (RMSE) between HR estimation and ground truth image (Liang et al., 2021; Zhang et al., 2024; Long et al., 2025), tend to produce overly smooth results (Ledig et al., 2017). To mitigate this limitation, generative SR (GSR) methods introduce impressive generative modeling techniques, e.g. generative adversarial networks (GANs) (Wang et al., 2018; 2021; Zhang et al., 2021) and diffusion-based models (Rombach et al., 2022; Yue et al., 2023; Wang et al., 2024b; Zhang et al., 2025), to obtain the

---

[*]Corresponding author

**Project page:** https://github.com/LabShuHangGU/TVQ-RAP

capability of prior distribution modeling, has been a thriving research topic due to its highly practical value in generating photo-realistic SR results.

Recently, besides the GAN-based and Diffusion-based generative modeling techniques, another category of generative visual modeling approaches, i.e. the vector-quantized variational autoencoder (VQ-VAE), has shown advantages in modeling accuracy and efficiency in image generation tasks (Van Den Oord et al., 2017; Esser et al., 2021; Ramesh et al., 2021; Lee et al., 2022; Tian et al., 2025). At the core of VQ-based model is a visual codebook, with which visual features are encoded as their corresponding nearest codebook items and visual prior is modeled by training codebook index predicting networks. Despite their great success in visual prior modeling, the existing VQ-based methods still suffer from the following two limitations. First, most of the existing VQ-based methods directly replace visual features with nearest codebook items, due to the richness and diversity of natural images, a large codebook is often required to fulfill the requirement of coding accuracy (see Figure 1 (a)). However, the incorporation of a large codebook not only introduces heavy memory footprint but also escalates training difficulty. Second, in the existing VQ-based methods, visual prior is captured by training the index predicting network with code-level supervision, i.e. minimizing cross-entropy

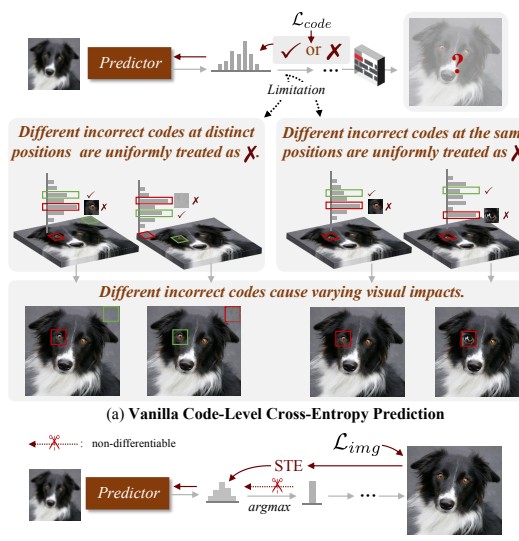

(a) **Vanilla Code-Level Cross-Entropy Prediction**

(b) **Image-Level Reconstruction aware Prediction**

Figure 2: (a) Code-level loss ignores the visual impacts caused by the predicting results and penalizes all non-ground-truth predictions equally. (b) Our reconstruction-aware training strategy guides the predictor according to the visual impacts introduced by different code predictions.

between predicted and target probability. This makes index prediction accuracy the primary optimization target, which in practice does not strictly align with image quality. As a result, such an indirect training paradigm ignores the different levels of reconstruction impacts introduced by different incorrect codes, penalizing all predictions that deviate from the ground-truth index even if the predicted code yields a visually plausible result (see Figure 2 (a)), which may cause optimization stagnation and ultimately result in sub-optimal prior modeling.

In this paper, we propose a novel VQ-based generative super resolution framework with **T**exture **V**ector-**Q**uantization (TVQ) and **R**econstruction **A**ware **P**rediction (RAP) strategies. Inspired from classical dictionary learning methods (Matsui et al., 2017; Zeyde et al., 2010; Gu et al., 2015), which remove low-frequency intensity component to improve the representation capability of dictionary, our TVQ strategy introduces visual texture codebook instead of vanilla codebook for predictive prior modeling. Concretely, we decompose image into the structure and the texture components; the structure component can be easily estimated by the LR input, and we only exploits texture codebook to encode the remaining texture features. Removing structure information could significantly reduce the diversity of feature space, therefore alleviating the coding error introduced by VQ and consequently improving prior modeling accuracy. An illustration of our Texture VQ strategy versus vanilla VQ paradigm can be found in Figure 1. Moreover, besides TVQ, another important innovation of our paper lies in our predictor training scheme. As we have discussed previously, most of the existing VQ-based methods (Van Den Oord et al., 2017; Esser et al., 2021; Zhou et al., 2022) train index predictor with code-level supervision which ignores the consequences of selective predicting errors, i.e. the final reconstruction error. While, we proposes a reconstruction aware training paradigm which directly exploits image-level reconstruction supervision for training the predictor. As illustrated in Figure 2, the predictor directly takes the quality of the reconstructed image into consideration, aligning the optimization target with image quality and is expected to better capture the visual prior for generating high-quality visual data. Building upon our proposed strategies, our proposed model is able to achieve state-of-the-art GSR results with less computational footprints.

The contributions of this paper are summarized as follows: **(i)** We present a tailored visual prior modeling framework for generative super-resolution, which takes inspiration from classical dictionary learning method and establish texture codebook to mitigate the encoding difficulty of highly complex visual signal. **(ii)** We propose an advanced training strategy for predictive visual prior modeling, which directly take the final image-level reconstruction accuracy instead of intermediate code-level predicting accuracy as target to train index predictor. **(iii)** We conduct comprehensive experiments on both synthetic and real-world datasets, our method is able to achieve state-of-the-art generative super-resolution results with less computational footprint; detailed ablation studies are also provided to validate the effectiveness of our innovations.

## 2 RELATED WORKS

### 2.1 VECTOR QUANTIZATION METHODS

The seminal VQ-VAE (Van Den Oord et al., 2017) introduced a learnable codebook to discretize continuous latent representations, providing a foundation for subsequent generative modeling approaches. Building upon this, VQGAN (Esser et al., 2021) incorporated adversarial losses during training, significantly improving the visual quality of reconstructed images. However, despite these advancements, the overall performance of VQ-based models remains limited by the expressive capacity of the codebook. To address this challenge, various strategies have been proposed to enhance the representational power of VQ models. These include RQVAE (Lee et al., 2022) with multi-stage recursive encoding for fine details, ViT-VQGAN (Yu et al., 2021) leveraging a larger codebook and lower compression ratio for higher fidelity, and MoVQ (Zheng et al., 2022) using multi-channel quantization to boost codebook expressiveness. While these techniques improve representation capacity, they often introduce trade-offs such as increased model complexity and computational cost. Moreover, existing VQ-based methods typically use per-code cross-entropy loss for code prediction, which limits the model's ability to capture the underlying distribution of visual data.

### 2.2 IMAGE SUPER-RESOLUTION

Image super-resolution (SR) is a longstanding ill-posed problem that remains a fundamental challenge in low-level vision. Traditional SR methods (Dong et al., 2012; Gu et al., 2015) rely on handcrafted priors and domain-specific knowledge to recover HR details. With the advent of deep learning, data-driven approaches have become dominant in the SR domain (Dong et al., 2015; Wang et al., 2020). Early SR methods (Liang et al., 2021; Zhang et al., 2024; Long et al., 2025) optimized pixel-wise losses (e.g., mean squared error) to achieve high PSNR, but often produced overly smooth results lacking realistic textures (Ledig et al., 2017). To address this limitation, photorealistic SR approaches adopt generative models such as GANs (Wang et al., 2018; 2021; Zhang et al., 2021) and diffusion models (Rombach et al., 2022; Yue et al., 2023; Wang et al., 2024b; Zhang et al., 2025; Wu et al., 2024; Yang et al., 2024) to better capture complex image priors, leading to the reconstruction of more natural and detailed textures. Despite significant advances, GAN-based methods continue to face challenges such as training instability and difficulty balancing perceptual quality with fidelity. Diffusion-based SR methods (Yue et al., 2023; Wang et al., 2024b; Zhang et al., 2025; Wu et al., 2024; Yang et al., 2024) often incur substantial computational costs during inference, which further diminishes their practicality for real-world applications.

### 2.3 VQ-BASED IMAGE SUPER-RESOLUTION

More recently, VQ-based super-resolution methods have emerged as promising alternatives by incorporating discrete generative priors to enhance reconstruction quality. However, since most of these methods inherit from VQ-based generative models, they face common limitations such as under-expressive codebooks and indirect optimization objectives, which lead to suboptimal predictors. For example, CodeFormer (Zhou et al., 2022) is specifically tailored for facial images, limiting its generalizability. FeMaSR (Chen et al., 2022) struggles with complex scenes, often yielding suboptimal restoration quality. AdaCode (Liu et al., 2023) introduces a multi-codebook quantization pipeline that increases both training and inference complexity. VARSR (Qu et al., 2025), despite its strong performance, depends on a complex multi-scale residual quantization mechanism and a large pretrained autoregressive predictor, and thus shares the common limitation of diffusion-based

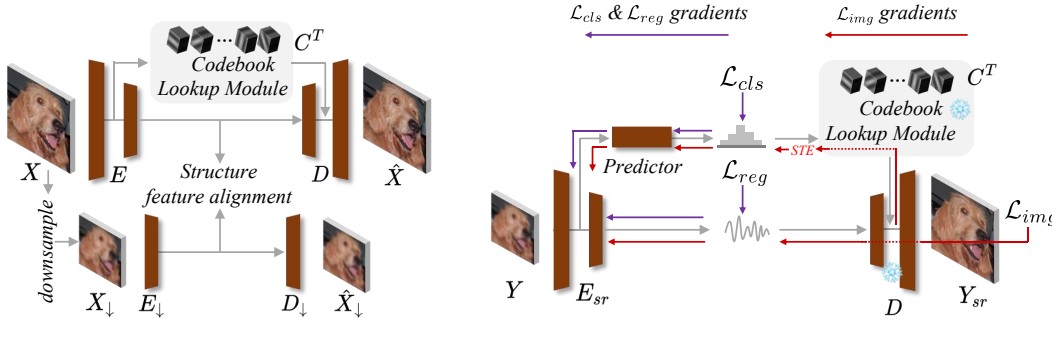

(a) **Texture Vector Quantization**  (b) **Reconstruction aware Prediction**

Figure 3: Overview of the proposed Texture Vector Quantization (TVQ) and Reconstruction Aware Prediction (RAP) strategies. **(a) Texture Vector Quantization**, we decompose the image into the structure and texture components, and only exploit codebook to generate discrete texture features; removing the structure component could significantly reduce the complexity of visual feature space, result in enhanced texture representation accuracy. **(b) Reconstruction Aware Prediction**, instead of training predictor through indirect code-level supervision, we introduce image-level supervision which take the reconstruction error lead by different predicting results into consideration; the predictor is trained to select codebook items for generating high-quality image details.

methods that use pretrained generative priors, such as high computational cost. In contrast, our proposed framework is explicitly designed to address these limitations. By introducing a texture-focused vector quantization scheme and incorporating image-level supervision in code prediction, our method significantly enhances representational capacity while enabling direct optimization for perceptual quality. Our method produces high-quality results while maintaining model efficiency.

## 3 METHODOLOGY

In this section, we present details of our proposed generative super-resolution method. We first introduce how high-quality images are decomposed into structure and texture components to facilitate learning a texture codebook for discrete texture encoding. Then, we describe our reconstruction aware prediction training strategy which uses straight-through estimator (STE) (Bengio et al., 2013) to train index predictor with image-level supervision.

### 3.1 IMAGE SEPARATION FOR TEXTURE VECTOR-QUANTIZATION

The VQ-based generative model (Van Den Oord et al., 2017; Esser et al., 2021; Ramesh et al., 2021; Lee et al., 2022; Tian et al., 2025; Zhou et al., 2022; Yu et al., 2021; Zheng et al., 2022) encodes continuous visual features with a learned codebook and trains a codebook index predicting network to capture visual prior. At the core of VQ-based model is a visual codebook which comprises typical visual features to encode continuous feature in a vector-quantization manner. The richness and diversity of natural images makes the latent space of visual feature a highly complex space, discrete representation with guaranteed reconstruction accuracy often relies on a large codebook with enormous number of typical features. In this paper, we study the generative super-resolution task, for which low-resolution information is available at the inference stage. The specific character of super-resolution task inspires us to remove the available structure information and only discretization the texture information for reducing the codebook complexity.

In order to decompose high-quality images into the structure and texture components, we train a multiscale autoencoder which extracts feature maps with two different resolutions, i.e. $\boldsymbol{F}^H \in \mathbb{R}^{C_H \times H_H \times W_H}$ and $\boldsymbol{F}^L \in \mathbb{R}^{C_L \times H_L \times W_L}$:

$$[\boldsymbol{F}^H, \boldsymbol{F}^L] = \mathbf{E}(\boldsymbol{X}), \tag{1}$$

where $\boldsymbol{X} \in \mathbb{R}^{3 \times H_I \times W_I}$ is the input high-quality image, $\mathbf{E}(\cdot)$ is the image encoder. We expect the low-resolution feature maps $\boldsymbol{F}^L$ and high-resolution feature maps $\boldsymbol{F}^H$ to encode the structure and

texture components, respectively. To achieve this goal, we generate a down-sampled low-resolution image $\boldsymbol{X}_\downarrow \in \mathbb{R}^{3 \times H_D \times W_D}$ and train another auto-encoder on the down-sampled image,

$$\boldsymbol{F}_\downarrow = \mathbf{E}_\downarrow(\boldsymbol{X}_\downarrow), \quad \hat{\boldsymbol{X}}_\downarrow = \mathbf{D}_\downarrow(\boldsymbol{F}_\downarrow); \tag{2}$$

where $\mathbf{E}_\downarrow(\cdot)$ and $\mathbf{D}_\downarrow(\cdot)$ are encoder and decoder for down-sampled image $\boldsymbol{X}_\downarrow$. Please note that $\boldsymbol{X}_\downarrow$ is an extreme low-resolution image which is smaller than the low-resolution image to be super-resolved in the testing phase, which means $\boldsymbol{F}_\downarrow$ only include basic structure information of the image. With the help of $\boldsymbol{F}_\downarrow$, we could disentangle basic structure information from $\boldsymbol{X}$ by aligning $\boldsymbol{F}^L$ with $\boldsymbol{F}_\downarrow$. Consequently, as $\boldsymbol{F}^L$ and vector-quantized version of $\boldsymbol{F}^H$ are required to reconstruct high-quality image, $\boldsymbol{F}^H$ is learned to represent the structure-removed texture information of $\boldsymbol{X}$. With separated image components, we introduce codebook to generate discrete texture representation via vector-quantization. Denote the texture codebook by $\boldsymbol{C}^T$, for each token in $\boldsymbol{F}^H$, it find nearest codebook item in $\boldsymbol{C}^T$ to establish vector-quantized texture feature $\boldsymbol{F}^{H-vq} = Lookup(\boldsymbol{F}^H, \boldsymbol{C}^T)$. Lastly, $\boldsymbol{F}^L$ and $\boldsymbol{F}^{H-vq}$ are combined to reconstruct the original high-quality image with decoder

$$\hat{\boldsymbol{X}} = \mathbf{D}(\boldsymbol{F}^{H-vq}, \boldsymbol{F}^L). \tag{3}$$

Following the commonly used VQ-GAN (Esser et al., 2021), we adopt MSE loss, perceptual loss and GAN loss to optimize the difference between $\boldsymbol{X}$ and $\hat{\boldsymbol{X}}$. The alignment between $\boldsymbol{F}^L$ and $\boldsymbol{F}_\downarrow$ is achieved by minimizing the their Euclidean distance. We use the same stop-gradient strategy as in (Van Den Oord et al., 2017; Esser et al., 2021; Ramesh et al., 2021; Lee et al., 2022; Tian et al., 2025) to deal with the back-propagation issue introduced by codebook. An illustration of our Image separation framework is shown in left part of Fig. 3. More implementation details can be found in the experimental section 4.1 and appendix B.

## 3.2 RECONSTRUCTION AWARE PREDICTION

With the above TVQ training, we are able to represent high-quality image as continuous maps $\boldsymbol{F}^L$ and discrete representation $\boldsymbol{F}^{H-vq}$, where $\boldsymbol{F}^L$ and $\boldsymbol{F}^{H-vq}$ can be combined to generate the original high-quality image. In the second stage of training, we aim to predict $\boldsymbol{F}^L$ and $\boldsymbol{F}^{H-vq}$ with the corresponding low-resolution input image $\boldsymbol{Y}$. Since $\boldsymbol{X}_\downarrow$ in TVQ training is with lower resolution than $\boldsymbol{Y}$, all the information in $\boldsymbol{F}^L$ can be easily regressed by $\boldsymbol{Y}$, the major difficulty of generative SR lies in predict $\boldsymbol{F}^{H-vq}$ from $\boldsymbol{Y}$. In vanilla VQ-based method, a probability predictor can be trained to predict the probability of codebook indexes with cross-entropy loss:

$$\mathcal{L}_{CE} = -\sum_i I_i^H \log(\hat{I}_i), \tag{4}$$

where $I^H$ are the target codes achieved by TVQ from HR image. Although that $\mathcal{L}_{CE}$ is able to guide the predictor to estimate correct code for reconstructing the high-quality image, it treats all the prediction errors equally and neglects the final reconstruction errors lead by different prediction choices. In order to reduce the reconstruction error, which is the ultimate target of super-resolution task, we introduce image-level supervision for training reconstruction aware index predictor. Considering the forward process of predictive image reconstruction, let us denote the one-hot index as:

$$\hat{I}_i^{one-hot} = OneHot(\hat{I}_i), \tag{5}$$

and the decoded texture feature is achieved by: $\hat{\boldsymbol{F}}_i^{H-vq} = \boldsymbol{C}^T(\hat{I}_i^{one-hot})$. We plug $\hat{\boldsymbol{F}}_i^{H-vq}$ into the pre-trained decoder in equation 3 to generate HR estimation, and backpropagate commonly used reconstruction losses including the MSE loss, the perceptual loss and GAN loss to train the index predictor. As the decoder is differentiable, the gradient can be easily back-propagated to $\hat{I}_i^{one-hot}$ through $\hat{\boldsymbol{F}}_i^{H-vq}$. To deal with the OneHot operator in Eq. equation 5, we reformulate $\hat{I}_i^{one-hot}$ as:

$$\hat{I}_i^{one-hot} = \hat{I}_i + (\hat{I}_i^{one-hot} - \hat{I}_i).detach \tag{6}$$

in the network. The above straight-through estimator (STE) trick has been widely used in various models. We use it to introduce image-level supervision for training code index predictor. In addition to predicting the code indices, we also need to extract structural information from the LR input. As $\boldsymbol{F}^L$ is continuous and $\boldsymbol{X}_\downarrow$ is with lower resolution than LR input, we simply MSE loss between $\hat{\boldsymbol{F}}^L$ and its corresponding $\boldsymbol{F}^L$ for supervision. More implementation details can be found in the experimental section 4.1 and appendix B.

Table 1: Quantitative results of models on *ImageNet-Test*. The best and second best results are highlighted in **bold** and underline. ("-N" behind the method represents the number of inference steps)

| Methods | PSNR↑ | SSIM↑ | LPIPS↓ | DISTS↓ | CLIPIQA↑ | MUSIQ↑ | MANIQA↑ | FID↓ |
|---|---|---|---|---|---|---|---|---|
| ESRGAN (Wang et al., 2018) | 20.67 | 0.448 | 0.485 | 0.3049 | 0.451 | 43.615 | 0.3212 | 73.02 |
| BSRGAN (Zhang et al., 2021) | 24.42 | 0.659 | 0.259 | 0.2207 | 0.581 | 54.697 | 0.3865 | 45.63 |
| SwinIR (Liang et al., 2021) | 23.99 | 0.667 | 0.238 | 0.2058 | 0.564 | 53.790 | 0.3882 | 35.73 |
| RealESRGAN (Wang et al., 2021) | 24.04 | 0.665 | 0.254 | 0.2174 | 0.523 | 52.538 | 0.3689 | 41.48 |
| FeMaSR (Chen et al., 2022) | 22.35 | 0.606 | 0.243 | 0.2089 | 0.662 | 55.930 | 0.4721 | 43.39 |
| AdaCode (Liu et al., 2023) | 23.30 | 0.626 | 0.237 | 0.2046 | 0.663 | 53.950 | 0.4171 | 40.59 |
| LDM-15 (Rombach et al., 2022) | 24.85 | 0.668 | 0.269 | 0.2101 | 0.510 | 46.639 | 0.3305 | 30.53 |
| ResShift-15 (Yue et al., 2023) | **24.94** | **0.674** | 0.237 | **0.1716** | 0.586 | 53.182 | 0.4191 | **19.53** |
| SinSR-1 (Wang et al., 2024b) | 24.70 | 0.663 | 0.218 | 0.1808 | 0.611 | 53.632 | 0.4161 | 25.58 |
| UPSR-5 (Zhang et al., 2025) | 23.77 | 0.630 | 0.246 | 0.2017 | 0.633 | 59.227 | 0.4591 | 37.92 |
| TVQ&RAP (Ours) | 22.49 | 0.603 | **0.210** | 0.1784 | **0.730** | **63.873** | **0.5530** | 26.57 |

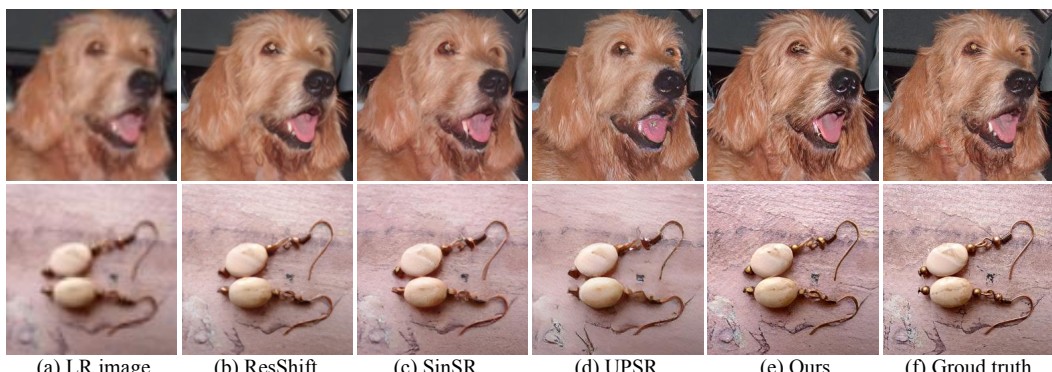

    (a) LR image      (b) ResShift      (c) SinSR      (d) UPSR      (e) Ours      (f) Groud truth

Figure 4: Qualitative comparison between different methods on *ImageNet-Test* dataset.

## 4 EXPERIMENTS

In this section, we conduct experiments to validate the effectivness of our proposed method. We firstly introduce our experimental settings, and then compare our method with recently proposed generative SR approaches. Lastly, a model analysis section is presented to validate the advantages of our proposed TVQ and RAP strategies.

### 4.1 EXPERIMENTAL SETTINGS

**Training details.** We follow the experimental settings of Yue et al. (2023) and train our method on the ImageNet training set (Deng et al., 2009). For training SR model with zooming factor 4, we utilize the degradation process in (Wang et al., 2021) to generate paired low-resolution (LR) and high-resolution (HR) images. The down-sampled image $X_\downarrow$ for structure disentangle is obtained by down-sampling the original image $X$ with a factor of 8. The spatial size of the structure components $F^L$ and texture components $F^H$ are 32 times and 8 times smaller than the size of HR image, i.e. $H_L = H_I/32$, $H_H = H_I/8$, with channel numbers 64 and 256, respectively. We introduce texture codebook with 1024 items, and conduct our TVQ training in Section 3.1 for 450K iterations with 512×512 images. As for the reconstruction aware prediction stage in Section 3.2, to reduce training time, we firstly train the the predictor with code-level cross-entropy loss for 300K iterations, and then finetune the predictor with image-level reconstruction aware training for another 10K iterations. Detailed network architectures can be found in appendix B.2.

**Testing details.** Following recent work (Yue et al., 2023; Wang et al., 2024b; Zhang et al., 2025), we evaluate our method on synthetic and real-world datasets. For the synthetic setting, we utilize the *ImageNet-Test* dataset following Yue et al. (2023), which contains 3,000 images randomly selected from the ImageNet validation set. Additionally, we adopt two real-world datasets, RealSR (Cai et al., 2019) and RealSet65 (Yue et al., 2023), to assess the generalizability of our model in practical scenarios. We report several commonly used quality measure metrics following previous works, including full-reference metrics: PSNR, SSIM (Wang et al., 2004), LPIPS (Zhang et al., 2018) and

Table 2: Quantitative results of models on two real-world datasets. The best and second best results are highlighted in **bold** and underline. Notably, as Real65 lacks ground-truth references, we report only non-reference metrics following (Yue et al., 2023; Wang et al., 2024b; Zhang et al., 2025).

| Methods | RealSR | | | | | | | RealSet65 | | | |
|---|---|---|---|---|---|---|---|---|---|---|---|
| | PSNR↑ | SSIM↑ | LPIPS↓ | CLIPIQA↑ | MUSIQ↑ | MANIQA↑ | NIQE↓ | CLIPIQA↑ | MUSIQ↑ | MANIQA↑ | NIQE↓ |
| ESRGAN (Wang et al., 2018) | **27.57** | 0.7742 | 0.4152 | 0.2362 | 29.037 | 0.2071 | 7.73 | 0.3739 | 42.366 | 0.3100 | 4.93 |
| BSRGAN (Zhang et al., 2021) | 26.51 | 0.7746 | **0.2685** | 0.5439 | 63.587 | 0.3702 | 4.65 | 0.6160 | 65.583 | 0.3888 | 4.58 |
| RealESRGAN (Wang et al., 2021) | 25.83 | 0.7726 | 0.2739 | 0.4923 | 59.849 | 0.3694 | 4.68 | 0.6081 | 64.125 | 0.3949 | 4.38 |
| FeMaSR (Chen et al., 2022) | 25.43 | 0.7540 | 0.2927 | 0.5598 | 58.774 | 0.3430 | 4.76 | 0.6821 | 64.416 | 0.4100 | 5.01 |
| AdaCode (Liu et al., 2023) | 26.26 | 0.7605 | 0.2773 | 0.6092 | 61.279 | 0.3567 | 4.26 | 0.6877 | 64.533 | 0.4043 | 4.65 |
| StableSR-200 (Wang et al., 2024a) | 26.19 | 0.7556 | 0.2806 | 0.4124 | 48.346 | 0.3021 | 5.87 | 0.4488 | 48.740 | 0.3097 | 5.75 |
| LDM-15 (Rombach et al., 2022) | 27.18 | **0.7853** | 0.3021 | 0.3748 | 48.698 | 0.2655 | 6.22 | 0.4313 | 48.602 | 0.2693 | 6.47 |
| ResShift-15 (Yue et al., 2023) | 26.80 | 0.7674 | 0.3411 | 0.5709 | 57.769 | 0.3691 | 5.93 | 0.6309 | 59.319 | 0.3916 | 5.96 |
| SinSR-1 (Wang et al., 2024b) | 26.01 | 0.7083 | 0.4015 | 0.6627 | 59.344 | 0.4058 | 6.26 | 0.7164 | 62.751 | 0.4358 | 5.94 |
| UPSR-5 (Zhang et al., 2025) | 26.44 | 0.7589 | 0.2871 | 0.6010 | 64.541 | 0.3828 | 4.02 | 0.6392 | 63.519 | 0.3931 | **4.23** |
| TVQ&RAP (Ours) | 24.71 | 0.7202 | 0.2944 | **0.6897** | 65.591 | **0.4337** | 3.97 | **0.7347** | 68.420 | **0.4814** | 4.34 |

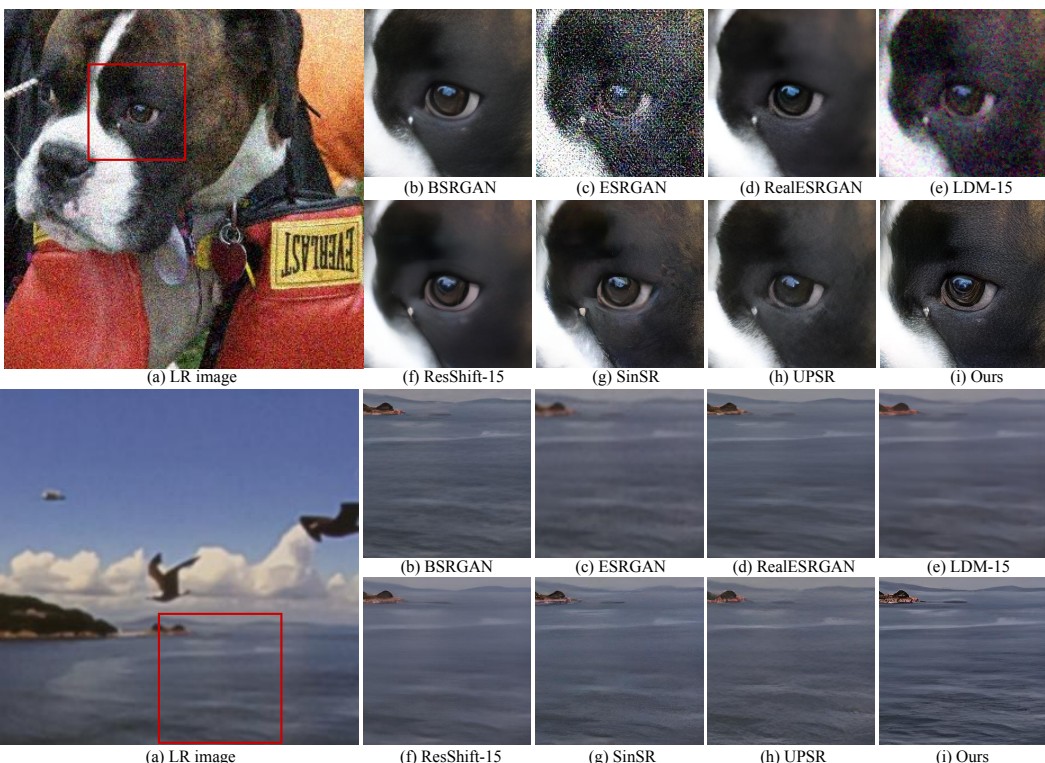

Figure 5: Qualitative comparison between different methods on two real-world datasets.

DISTS (Ding et al., 2020), and no-reference metrics: FID (Heusel et al., 2017), NIQE (Mittal et al., 2012), CLIPIQA (Wang et al., 2023), MANIQA (Yang et al., 2022), and MUSIQ (Ke et al., 2021).

## 4.2 COMPARISON WITH OTHER GENERATIVE SUPER-RESOLUTION METHODS

We benchmark our approach against several representative SR methods: ESRGAN (Wang et al., 2018), BSRGAN (Zhang et al., 2021), SwinIR (Liang et al., 2021), RealESRGAN (Wang et al., 2021), FeMaSR (Chen et al., 2022), AdaCode (Liu et al., 2023), StableSR (Wang et al., 2024a), LDM (Rombach et al., 2022), ResShift (Yue et al., 2023), SinSR (Wang et al., 2024b) and UPSR (Zhang et al., 2025). Table 1 and Table 2 report quantitative results on the synthetic ImageNet-Test and two real-world validation sets, respectively. On the ImageNet-Test dataset, our method attains the highest scores for both reference-based and no-reference perceptual metrics, while incurring minimal PSNR/SSIM degradation compared to the best models. On real-world datasets, our method either the best or the second best performance across the no-reference metrics. Figure 4 and Figure 5

present visual examples on synthetic datasets and real-world datasets: our reconstructions exhibit richer details and more realistic textures, with virtually less artifacts. More comparison and visual examples are provided in appendix C, D, F, I.

In addition to superior super-resolution results, another important advantage of our model lies in its efficiency. In Table 3, we compare the runtime and the number of parameters of several recently proposed generative super-resolution methods, including two VQ-based methods and several sota diffusion-based methods. Following Yue et al. (2023); Wang et al. (2024b); Zhang et al. (2025), we report runtime (ms), params (MB), and additionally several perceptual metrics on the *ImageNet-Test*

set from Table 1 for ease of comparison. As shown in Table 3, our predictive method is able to deliver photorealistic GSR results with high efficiency. In comparison to state-of-the-art multi-step diffusion based methods, i.e. ResShift-15 (Yue et al., 2023) and UPSR-5 (Zhang et al., 2025), our model is able to obtain comparable or better results with 5.5% and 16.5% of their runtime; in comparison with distilled one-step method SinSR-1 (Wang et al., 2024b), our method could utilize less than 60% of its runtime to obtain better GSR results. In terms of parameter count, our model also demonstrates competitiveness compared with competing methods.

Table 3: We compare runtime efficiency and perceptual performance with state-of-the-art methods. All models are evaluated on 64×64 input images using a single RTX 3090 GPU. The best results are highlighted in **bold**.

| Methods | Runtime | Params | LPIPS↓ | MUSIQ↑ | CLIPIQA↑ |
|---|---|---|---|---|---|
| FeMaSR | 57ms | **34M** | 0.243 | 55.930 | 0.662 |
| AdaCode | 104ms | 57M | 0.237 | 53.950 | 0.663 |
| LDM-15 | 223ms | 114M | 0.2685 | 46.639 | 0.510 |
| ResShift-15 | 689ms | 119M | 0.2371 | 53.128 | 0.586 |
| SinSR-1 | 65ms | 119M | 0.2183 | 52.632 | 0.611 |
| UPSR-5 | 230ms | 119M | 0.2460 | 59.227 | 0.633 |
| Ours | **38**ms | 57M | **0.2101** | **63.873** | **0.730** |

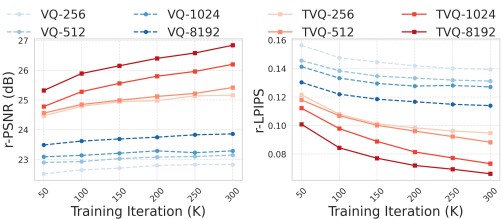

Figure 6: Comparisons between vanilla codebook and our proposed texture codebook.

Table 4: A comparison between Vanilla Codebook and Texture Codebook. Evaluation is conducted on *ImageNet-Test*, where 'r-' denotes reconstruction metrics. Experimental details can be found in Section 4.3.

| Methods | r-PSNR↑ | r-LPIPS↓ | r-FID↓ | PSNR↑ | LPIPS↓ | FID↓ |
|---|---|---|---|---|---|---|
| VQ | 23.29 | 0.1271 | 12.81 | 22.87 | 0.2707 | 44.54 |
| TVQ | **26.20** | **0.0733** | **6.49** | **24.10** | **0.2216** | **33.23** |

## 4.3 MODEL ANALYSIS

In this part, we present detailed ablation studies to analyzes the advantages of our proposed **Texture Vector Quantization** (TVQ) and **Reconstruction Aware Prediction** (RAP) strategies.

**Effect of Texture Vector-Quantization.** To evaluate the effectiveness of the proposed texture vector quantization, we conduct ablation studies with a lightweight variant of our architecture. We compare our method against a vanilla baseline with the structure branch removed. A series of experiments examining performance across different codebook sizes and training iterations are presented. As shown in Figure 6, our method consistently achieves better performance under the same codebook size and training iterations. Moreover, it outperforms competing methods even with smaller codebooks and fewer training iterations. Notably, TVQ-256 at 100k iterations surpasses VQ-8192 at 300k, highlighting that our approach enables more efficient codebook representation, thereby enhancing prior modeling capability. To further

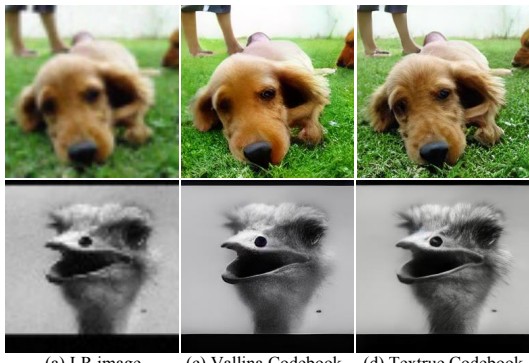

(a) LR image    (c) Vallina Codebook    (d) Textrue Codebook

Figure 7: Visual comparisons between the super-resolution results with vanilla codebook and our proposed texture codebook. Experimental details can be found in Section 4.3.

evaluate the benefit of the stronger prior for SR, we perform ablation studies on the SR task, comparing our method with the vanilla baseline under a codebook size of 1024. For both models, we use only the code-level loss to better isolate and verify the effectiveness of the texture codebook. Both reconstruction and SR performance are evaluated on the *ImageNet-Test* dataset. As shown in Table 4, the texture vector quantization substantially outperforms the vanilla baseline by a large margin, demonstrating its superior representational capacity, which is highly beneficial for SR. Two visual examples are provided in Figure 7. The model with texture codebook could generate photorealistic images with vivid textures. The above quantitative and qualitative advantages of texture codebook over the vanilla codebook clearly validated our idea of texture vector quantization.

Table 5: A comparison between Code-Level supervision only and the integration of Image-Level supervision on *ImageNet-Test*. Experimental details can be found in Section 4.3.

| Method | Accuracy↑ | DISTS↓ | LPIPS↓ | FID↓ | CLIPIQA↑ | MUSIQ↑ | MANIQ↑ |
|---|---|---|---|---|---|---|---|
| Code-level supervision only | **6.8%** | 0.1935 | 0.2159 | 32.876 | 0.6971 | 61.687 | 0.5303 |
| +Image-level supervision | 4.4% | **0.1784** | **0.2101** | **26.567** | **0.7304** | **63.873** | **0.5530** |

**Effect of Reconstruction Aware Prediction.** To assess the effectiveness of the proposed Reconstruction-Aware Prediction strategy, we compare the super-resolution results of two training regimes: (1) models trained solely with code-level cross-entropy loss, and (2) models further fine-tuned using image-level supervision. As reported in Table 5, while code-level supervision achieves better index accuracy, incorporating image-level supervision yields substantial gains in both perceptual quality and structural fidelity. This indicates that code-level loss targeting index accuracy does not always directly correlate with image quality, whereas the proposed reconstruction aware prediction strategy better aligns with the goal of high-quality image reconstruction and thereby significantly enhances GSR results. Figure 8 presents representative visual results from our ablation study, further corroborating this conclusion. Models trained with

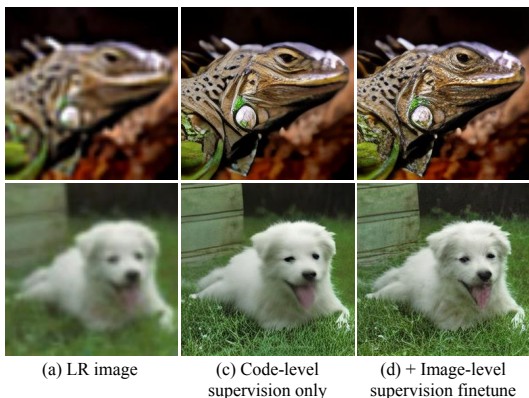

(a) LR image | (c) Code-level supervision only | (d) + Image-level supervision finetune

Figure 8: Visual comparisons between code-level supervision and our proposed reconstruction aware image-level supervision. Experimental details can be found in Section 4.3.

image-level supervision produce more subtle detailed textures that are often lost in code-only training. These improvements are especially pronounced in regions with complex patterns or high-frequency details. The superior GSR results achieved by Reconstruction aware prediction suggest that image-level supervision provides strong and explicit gradient signals to the code prediction network, significantly enhances the predictor's ability to generate high quality reconstruction results.

Collectively, our ablation study clearly validate the effectiveness of the proposed TVQ strategy and RAP strategy. The TVQ strategy enhances the representational capacity by focusing on texture details, while the RAP strategy improves the predictor's ability to generate perceptually accurate reconstructions through direct optimization with reconstruction-aware supervision. With the help of the two strategies, we are able to obtain state-of-the-art generative super-resolution results with less computatinal footprint. More ablation study are provided in appendix E, G.

## 4.4 FEATURE MAP RESOLUTION SELECTION IN OUR ARCHITECTURE

As discussed in Section 3.1, we represent an image using two components. The resolutions of the structure and texture components are downsampled by factors of 32× and 8×, relative to the HR image. For the texture branch, we follow the prior VQ-based super-resolution method (Zhou et al., 2022), adopting an 8× downsampling strategy. This choice balances detailed representation and computational efficiency. For the structure branch, we empirically adopt a larger downsampling factor of 32×, motivated by the observation that structures can be effectively captured at coarser resolutions.

To investigate the impact of feature map resolution on SR performance, we conduct a focused study using a lightweight variant. Specifically, we perform experiments with downsampling factors of 128×, 64×, 32×, 16×, and 8× relative to the HR image. As shown in Table 9, although 16× and 8× downsampling achieve better reconstruction performance, the 32× configuration yields the best results in SR. We attribute the poorer SR performance at 16× and 8× to the excessively large feature maps, which make it difficult—despite the use of alignment loss—to fully suppress texture information leakage through the structure branch. On the other hand, compared to 128× and 64× downsampling, the 32× setting retains relatively complete structural information, which is beneficial for effective decoupling of structure and texture features.

Table 6: A a abaltion study on different downsampling rates for the structure branch in our architecture. Evaluation is conducted on *ImageNet-Test*, where 'r-' denotes reconstruction metrics.

| Methods | r-PSNR↑ | r-LPIPS↓ | r-DISTS ↓ | r-FID↓ | PSNR↑ | LPIPS↓ | DISTS↓ | FID↓ |
|---------|---------|----------|-----------|--------|-------|--------|--------|------|
| 128× | 24.10 | 0.1196 | 0.1210 | 13.03 | 23.50 | 0.2279 | 0.1986 | 35.00 |
| 64× | 24.70 | 0.1046 | 0.1101 | 10.54 | 23.70 | 0.2241 | 0.1969 | 34.08 |
| 16× | 27.65 | 0.0525 | 0.0629 | 4.84 | 24.80 | 0.2594 | 0.2424 | 44.60 |
| 8× | **33.43** | **0.0147** | **0.0239** | **1.78** | **24.57** | 0.4285 | 0.3425 | 72.57 |
| 32× | 25.26 | 0.0898 | 0.0988 | 8.76 | 24.01 | **0.2220** | **0.1968** | **33.23** |

## 5    CONCLUSION

In this paper, we propose TVQ&RAP, a VQ-based method for generative super-resolution. To reduce the quantization error introduced by visual feature vector quantization, we decompose the image into structure and texture components and propose a texture vector-quantization (TVQ) strategy which introduce texture codebook to mitigate the difficulty in discrete visual representation. Furthermore, in order to better training the prediction network, we suggest a reconstruction aware prediction (RAP) strategy which utilizes the final reconstruction error to train code index predictor in an end-to-end manner. With reduced difficulty in discrete visual representation and enhanced capability in detail reconstruction, we combine our proposed TVQ and RAP to establish a novel generative super-resolution framework. Extensive experimental results on synthetic and real-world datasets are provided to evaluatet the proposed method. Our model is able to achieve state-of-the-art generative super-resolution results with less computational footprint. Detailed ablation analysis are also provided to validate the effectiveness of the proposed TVQ and RAP strategies.

## REPRODUCIBILITY STATEMENT

We provide detailed hyperparameter settings in Section 4.1 and Appendix B. To further facilitate reproducibility, we will release our implementation and trained model checkpoints, enabling the reported results to be reproduced.

## THE USE OF LARGE LANGUAGE MODELS (LLMS)

We used large language models (LLMs) to aid in polishing the writing. Specifically, LLMs were employed to improve grammar, clarity, and readability of the manuscript. No part of the research ideation, methodological design, or experimental analysis relied on LLMs.

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

## A   THE USE OF LARGE LANGUAGE MODELS (LLMs)

We used large language models (LLMs) to aid in polishing the writing. Specifically, LLMs were employed to improve grammar, clarity, and readability of the manuscript. No part of the research ideation, methodological design, or experimental analysis relied on LLMs.

## B   IMPLEMENTATION DETAILS

### B.1   TRAINING DETAILS

As discuss in section 3.1, to supervise the multiscale tokenizer, following VQ-GAN (Esser et al., 2021), we adopt a compound loss including: codebook loss $\mathcal{L}_{\text{codebook}}$, commit loss $\mathcal{L}_{\text{commit}}$, MSE loss $\mathcal{L}_{\text{mse}}$, perceptual loss $\mathcal{L}_{\text{per}}$ (Johnson et al., 2016; Zhang et al., 2018), and adversarial loss $\mathcal{L}_{\text{adv}}$ (Esser et al., 2021). The overall loss function is formulated as:

$$\mathcal{L} = \mathcal{L}_{\text{codebook}} + \mathcal{L}_{\text{commit}} + \mathcal{L}_{\text{mse}}(\hat{\boldsymbol{X}}) + \mathcal{L}_{\text{per}}(\hat{\boldsymbol{X}}) + \lambda_{\text{adv}} \cdot \mathcal{L}_{\text{adv}}(\hat{\boldsymbol{X}}), \tag{7}$$

where $\lambda_{\text{adv}}$ is a weighting factor, set to 0.75 empirically in our training. For the reconstruction task of the tokenizer applied to $\boldsymbol{X}_{\downarrow}$, our objective is not to generate a visually perfect image, but rather to extract a meaningful feature representation that captures the basic structure information. Hence, we employ a basic MSE loss:

$$\mathcal{L} = \mathcal{L}_{\text{mse}}(\hat{\boldsymbol{X}}_{\downarrow}). \tag{8}$$

As discuss in section 3.2, we supervise the super resolution pipeline using both code-level and image-level objectives. Specifically, the code-level ground truths: $\hat{\boldsymbol{F}}^L$ and $I^H$, are obtained by feeding the corresponding high-resolution image $\boldsymbol{X}$ into our pretrained reconstruction network. The code-level loss consists of a MSE loss for regression and a cross-entropy loss for classification:

$$\mathcal{L}_{\text{code}} = \|\boldsymbol{F}^L - \hat{\boldsymbol{F}}^L\|_2^2 + \lambda_{\text{CE}} \cdot (-\sum\nolimits_i I_i^H \, \log(\hat{I}_i)). \tag{9}$$

where $\lambda_{\text{CE}}$ is a weighting factor that balances the two losses, empirically set to 0.5 in our training. For the image-level supervision, we adopt the same loss formulation as Equation 7.

## B.2 NETWORK ARCHITECTURES

Following prior work (Esser et al., 2021; Zhou et al., 2022; Liu et al., 2023), we design a multiscale VQ-tokenizer composed of residual blocks (He et al., 2016) and attention layers (Vaswani et al., 2017; Liu et al., 2021; Liang et al., 2021). The tokenizer encodes the image into token maps at two spatial resolutions, with downsampling factors of 8 and 32, respectively. The texture codebook contains $N = 1024$ entries. The predictor is implemented using 12 Swin-Attention blocks. This modular design ensures efficiency while maintaining strong representational capacity.

## C COMPARISONS TO PRETRAINING-BASED SR METHODS.

Table 7: Comparisons with pretraining-based SR methods on RealSR.

| Method | Runtime | Params | Memory | LPIPS↓ | DISTS↓ | FID↓ | MANIQA↑ | CLIPIQA↑ | NIQE↓ |
|---|---|---|---|---|---|---|---|---|---|
| SeeSR (Wu et al., 2024) | 5740ms | 2524M | 8.8G | **0.2806** | **0.1781** | 55.58 | 0.6122 | 0.6824 | 4.54 |
| VARSR (Qu et al., 2025) | 322ms | 1102M | 11.1G | 0.3232 | 0.2025 | 61.53 | **0.6176** | **0.7020** | 4.49 |
| Ours | **110**ms | **57**M | **1.2**G | 0.2944 | 0.1793 | **54.97** | 0.5807 | 0.6897 | **3.97** |

Although methods based on pretrained generative models have demonstrated impressive performance, their dependence on large, fixed backbones restricts flexibility—particularly when adapting to lightweight architectures. This significantly limits their suitability for deployment in real-world, resource-constrained environments. Moreover, such methods typically require massive model sizes and incur substantial inference costs, placing them on a distinct path from our proposed approach. Nevertheless, for completeness, we include comparisons with some state-of-arts pretrained-based methods. Since our previous method of calculating MANIQA was different from (Wu et al., 2024; Qu et al., 2025), we followed their testing approach and conducted the tests again. We evaluate quality metrics on uncropped image and evaluate the Runtime and the Memory on 128×128 inputs using a single RTX 4090 GPU. As reported in Table 7, our method achieves competitive performance while using significantly fewer parameters and requiring much less inference time. Specifically, SeeSR incurs a significant computational overhead, with 52× inference time and 44× parameters, whereas VARSR also exhibits high resource demands, requiring 3× the inference time and 19× the parameters.

## D EXPERIMENTS ON HIGH-RESOLUTION SCENARIOS

To evaluate our approach under high-resolution settings, we conducted additional experiments on the DRealSR dataset, which contains real-world 4K–5K images. Table 8 shows the superior performance of our method compared to recent sota methods.

Table 8: Comparisons on DRealSR.

| Method | CLIPIQA ↑ | MUSIQ ↑ | MANIQA ↑ | NIQE ↓ |
|---|---|---|---|---|
| SinSR | 0.6953 | 30.789 | 0.3589 | 5.79 |
| UPSR | 0.5319 | 33.060 | 0.3220 | 4.50 |
| Ours | **0.7377** | **34.102** | **0.4086** | **3.89** |

## E FEATURE MAP RESOLUTION SELECTION IN OUR ARCHITECTURE

As discussed in Section 3.1, we represent an image using two components. The resolutions of the structure and texture components are downsampled by factors of 32× and 8×, relative to the HR image. For the texture branch, we follow the prior VQ-based super-resolution method (Zhou et al., 2022), adopting an 8× downsampling strategy. This choice balances detailed representation and computational efficiency. For the structure branch, we empirically adopt a larger downsampling factor of 32×, motivated by the observation that structures can be effectively captured at coarser resolutions.

To investigate the impact of feature map resolution on SR performance, we conduct a focused study using a lightweight variant. Specifically, we perform experiments with downsampling factors of 128×, 64×, 32×, 16×, and 8× relative to the HR image. As shown in Table 9, although 16× and 8× downsampling achieve better reconstruction performance, the 32× configuration yields the best results in SR. We attribute the poorer SR performance at 16× and 8× to the excessively large feature maps, which make it difficult—despite the use of alignment loss—to fully suppress texture information leakage through the structure branch. On the other hand, compared to 128× and 64× downsampling,

the 32× setting retains relatively complete structural information, which is beneficial for effective decoupling of structure and texture features.

Table 9: A a abaltion study on different downsampling rates for the structure branch in our architecture. Evaluation is conducted on *ImageNet-Test*, where 'r-' denotes reconstruction metrics.

| Methods | r-PSNR↑ | r-LPIPS↓ | r-DISTS ↓ | r-FID↓ | PSNR↑ | LPIPS↓ | DISTS↓ | FID↓ |
|---------|---------|----------|-----------|--------|-------|--------|--------|------|
| 128× | 24.10 | 0.1196 | 0.1210 | 13.03 | 23.50 | 0.2279 | 0.1986 | 35.00 |
| 64× | 24.70 | 0.1046 | 0.1101 | 10.54 | 23.70 | 0.2241 | 0.1969 | 34.08 |
| 16× | 27.65 | 0.0525 | 0.0629 | 4.84 | 24.80 | 0.2594 | 0.2424 | 44.60 |
| 8× | **33.43** | **0.0147** | **0.0239** | **1.78** | **24.57** | 0.4285 | 0.3425 | 72.57 |
| 32× | 25.26 | 0.0898 | 0.0988 | 8.76 | 24.01 | **0.2220** | **0.1968** | **33.23** |

## F    SUBJECTIVE EVALUATION

Following the evaluation protocol of VARSR (Qu et al., 2025), we conduct a user study with 15 participants. Our method was compared against five representative ISR baselines (BSRGAN (Zhang et al., 2021), Real-ESRGAN (Wang et al., 2021), Resshift (Yue et al., 2023), UPSR (Zhang et al., 2025), and SinSR (Wang et al., 2024b)), using 90 images selected from three datasets: ImageNet-Test, RealSR, and RealSet65 (the first 30 images from each). For each image, participants were asked to select the best restoration among the six methods. This resulted in a total of 1350 responses (15 participants × 90 images). The results in Table 10 demonstrate that our method achieves the highest user preference rate (48.8%), significantly outperforming other approaches.

Table 10: Results of User Study

| Method | BSRGAN | Real-ESRGAN | Resshift | SinSR | UPSR | Ours |
|--------|--------|-------------|----------|-------|------|------|
| Preference (%) | 0.0% | 7.7% | 10.0% | 21.1% | 12.2% | 48.8% |

## G    WHAT IS REPRESENTED IN TWO FEATURE MAP

To show analysis that what is represented in $\boldsymbol{F}^L$ and $\boldsymbol{F}^H$, we conducted an additional analysis by passing $\boldsymbol{F}^L$ and $\boldsymbol{F}^H$ separately through the decoder to obtain corresponding reconstructions. The qualitative results in Figure 9 clearly show that: the $\boldsymbol{F}^L$-only reconstructions preserve coarse structures and smooth areas, and the $\boldsymbol{F}^H$-only reconstructions retain high-frequency textures without clear structural outlines. This aligns with our ideal of feature decomposition.

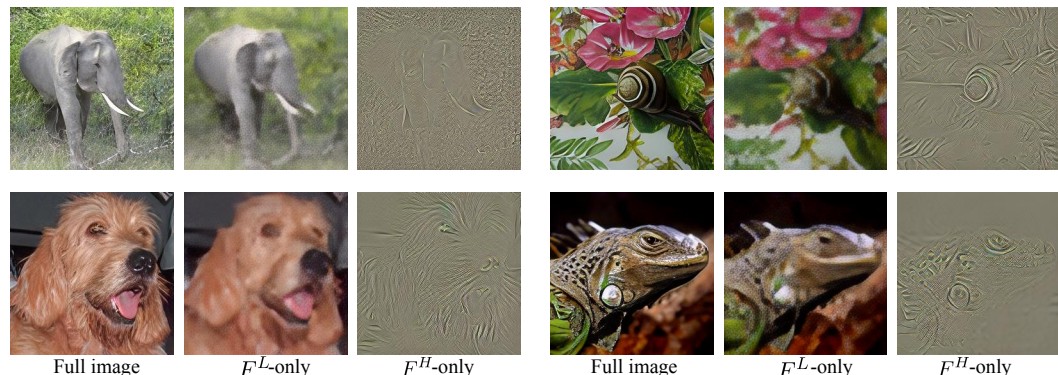

| Full image | $F^L$-only | $F^H$-only | Full image | $F^L$-only | $F^H$-only |

Figure 9: Qualitative comparison between different methods on *ImageNet-Test* dataset.

## H    QUANTITATIVE ANALYSIS OF REDUCED FEATURE-SPACE REDUNDANCY

To support our claim that TVQ reduces redundancy in the quantized feature space, the current manuscript provides indirect evidence: under the same (or even smaller) codebook sizes, TVQ

Table 11: K-means clustering distortion (lower is better) on latent features under the same number of clusters $K$.

| $K$ (clusters) | Distortion ↓ (VQ latent) | Distortion ↓ (TVQ latent) | Relative reduction |
|---|---|---|---|
| 64 | 16.53 | 12.96 | 21.6% |
| 128 | 14.19 | 10.53 | 25.8% |
| 256 | 11.93 | 8.24 | 30.9% |
| 512 | 9.55 | 6.08 | 36.4% |

Table 12: Covariance statistics of latent features. $\Sigma_{\text{reg}}$ denotes a diagonally regularized covariance matrix.

| Model | Total variance $\text{tr}(\Sigma) \downarrow$ | $\log \det(\Sigma_{\text{reg}}) \downarrow$ |
|---|---|---|
| VQ latent | $3.24 \times 10^1$ | $-6.37 \times 10^1$ |
| TVQ latent | $2.16 \times 10^1$ | $-1.11 \times 10^2$ |

achieves lower reconstruction/quantization error than vanilla VQ (Sec. 4.3). The intuition is that, by removing structural information already available from the LR input, TVQ narrows the content that must be represented by the discrete codebook, so the latent space becomes easier to approximate with finite capacity.

In the revision, we further provide a direct quantitative characterization of the latent space compactness from two complementary perspectives.

**(a) K-means clustering distortion under fixed capacity.** We treat latent vectors as point clouds and run $k$-means clustering on (i) the vanilla VQ latent and (ii) the TVQ texture latent using the same number of clusters $K$. We report the clustering distortion as the mean squared distance from each feature to its nearest cluster center. As shown in Table 11, TVQ consistently yields lower distortion than vanilla VQ across $K \in \{64, 128, 256, 512\}$, and the relative reduction increases with $K$ (from 21.6% to 36.4%). For instance, at $K = 512$ the distortion decreases from 9.55 to 6.08. This indicates that, for a *fixed discrete capacity* (same number of clusters/codewords), the TVQ texture space can be approximated substantially more accurately, implying a simpler and less redundant feature distribution.

**(b) Covariance structure and distribution volume.** We also compute the empirical covariance matrix $\Sigma$ of latent features for both models. Compared to vanilla VQ, the TVQ texture latent exhibits a lower total variance (i.e., $\text{tr}(\Sigma)$ reduced by $\sim 33\%$) and a smaller generalized variance, measured by $\log \det(\Sigma_{\text{reg}})$ with a small diagonal regularization. The results in Table 12 suggest that, in the same ambient dimensionality, the TVQ texture distribution concentrates in a more compact region of feature space, with both reduced overall variance and reduced effective volume.

**Summary.** Together with the lower quantization/reconstruction error under the same codebook size reported in Sec. 4.3, the clustering and covariance analyses above provide direct evidence that removing LR-predictable structure reduces redundancy in the quantized feature space. Consequently, the TVQ texture latent is more compact and easier to partition with a finite codebook, which aligns with our motivation for the proposed design.

## I   VISUAL COMPARISON

We provide more visual examples of our method compared with recent state-of-the-art methods on *ImageNet-Test* and real- world datasets. Visual examples are shown in Figure 10, 11, 12, 13, and 14.

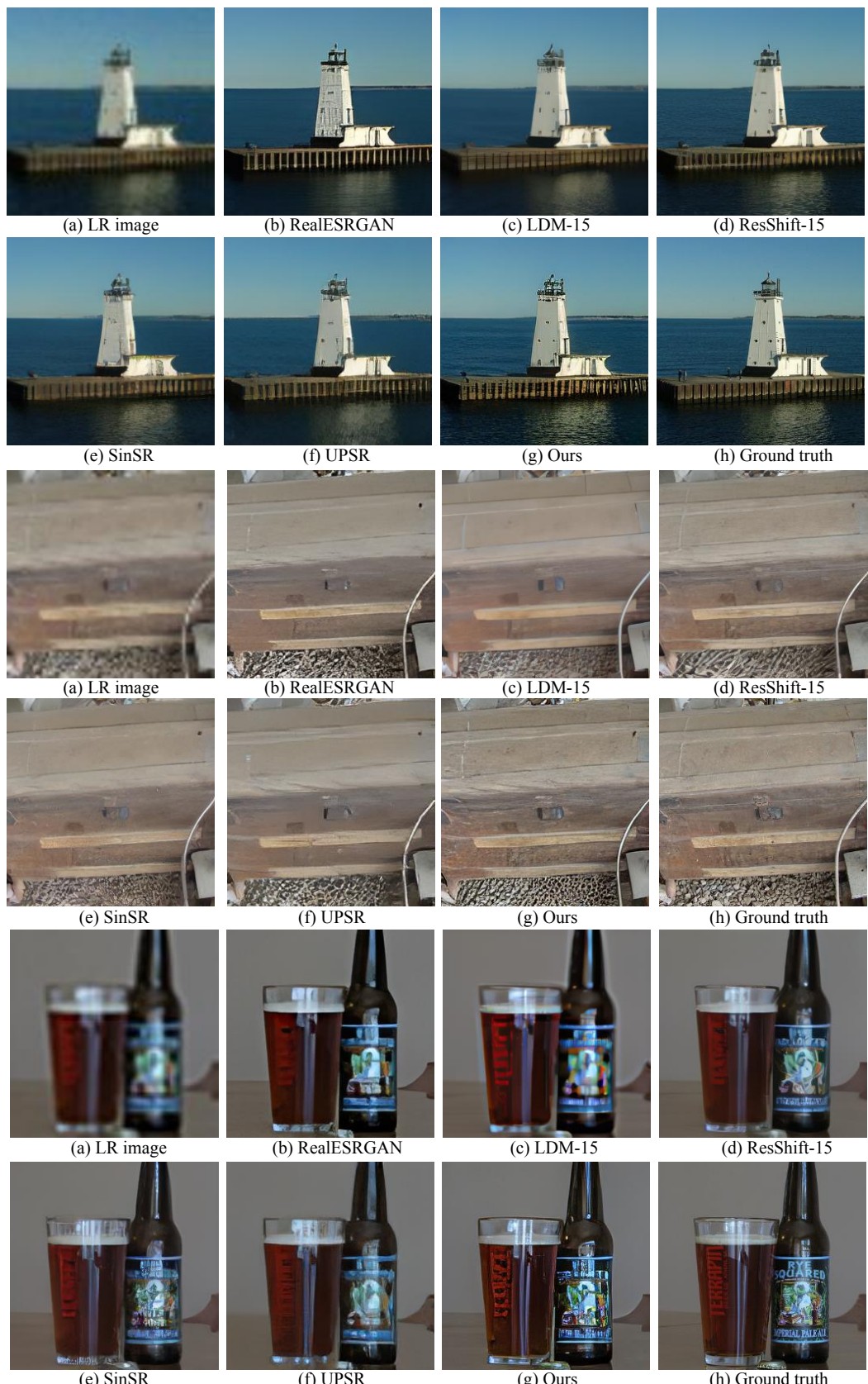

Figure 10: Qualitative comparison between different methods on *ImageNet-Test* dataset.

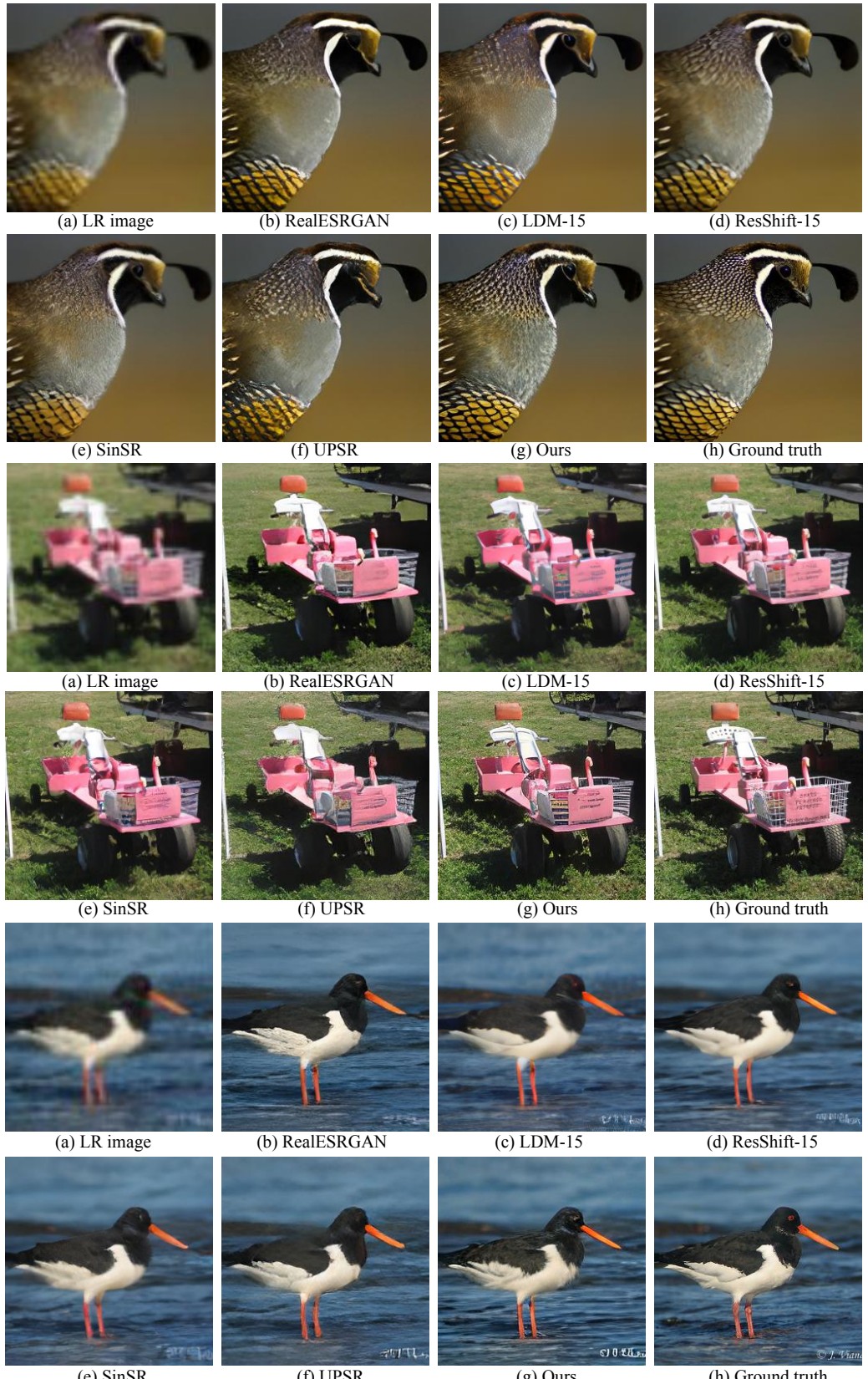

Figure 11: Qualitative comparison between different methods on *ImageNet-Test* dataset.

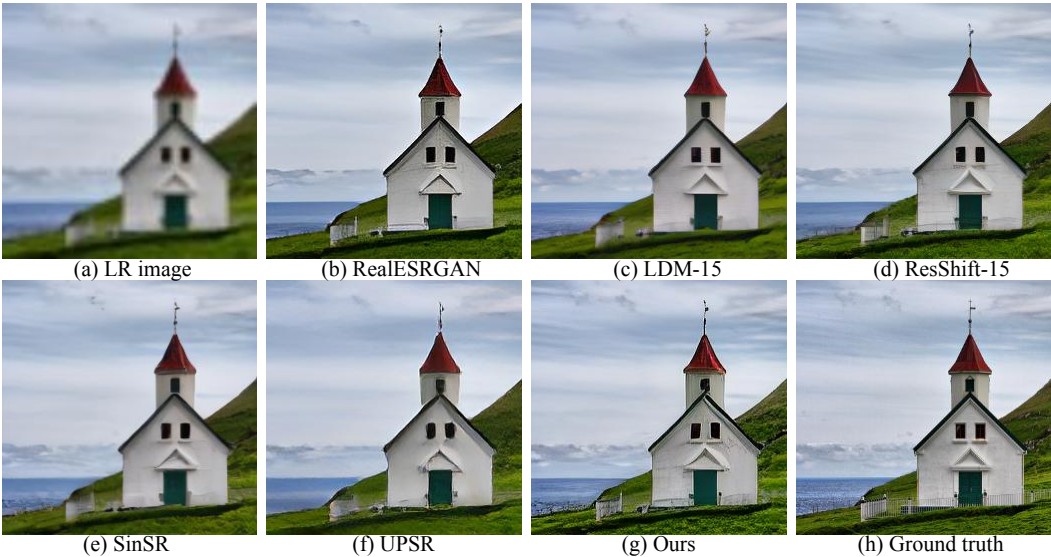

Figure 12: Qualitative comparison between different methods on *ImageNet-Test* dataset.

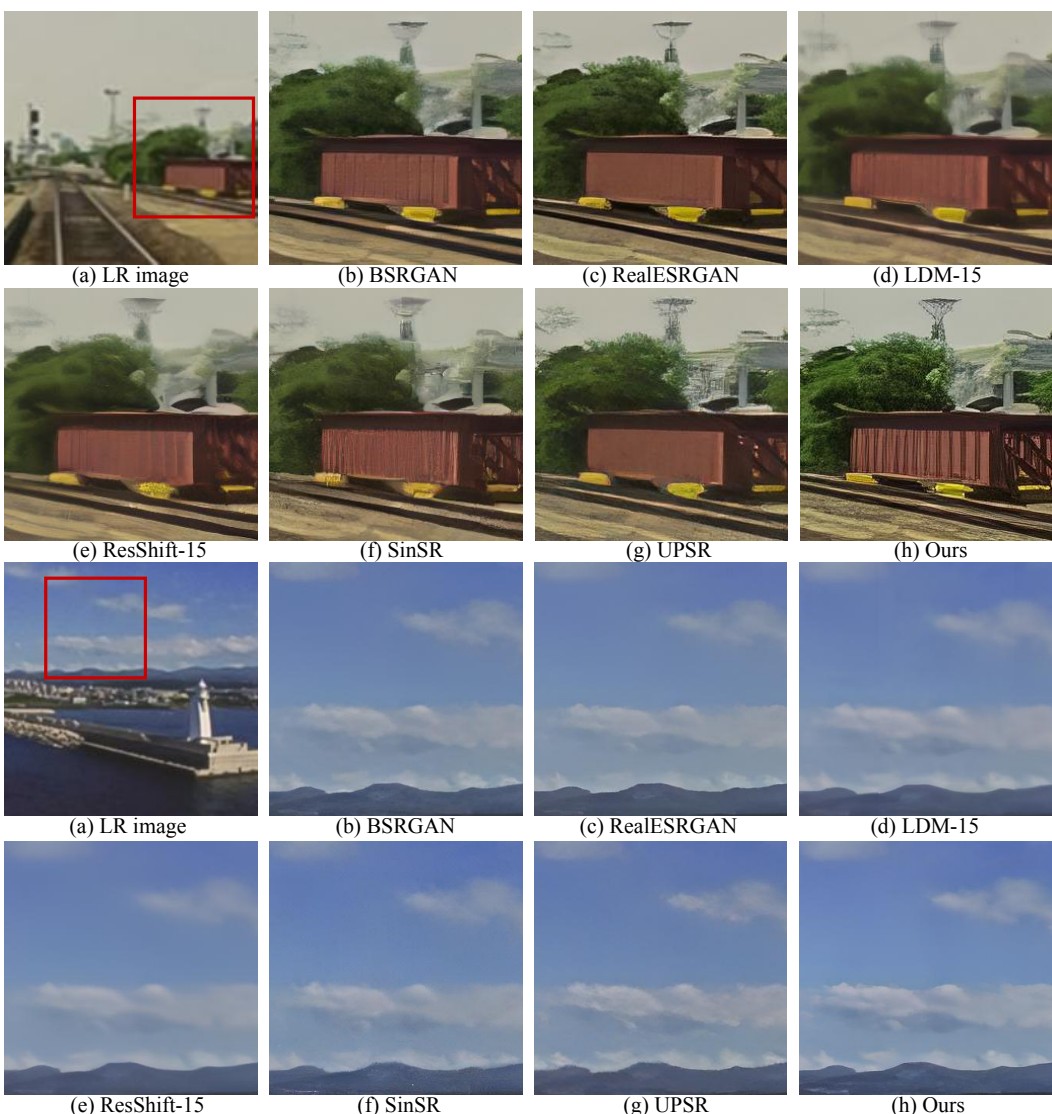

Figure 13: Qualitative comparison between different methods on two real-world datasets.

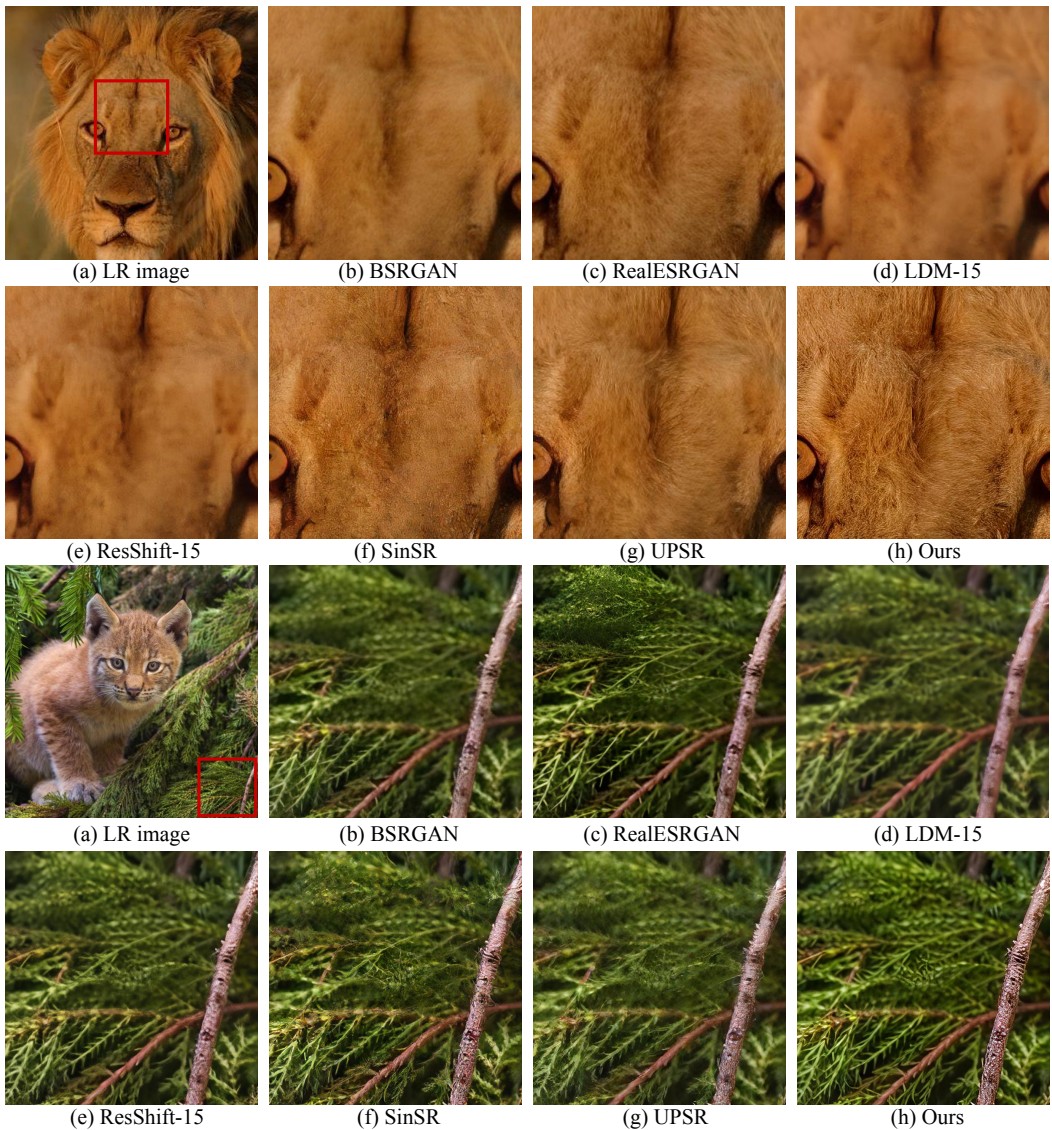

Figure 14: Qualitative comparison between different methods on two real-world datasets.

