# OpenReview forum: "Texture Vector-Quantization and Reconstruction Aware Prediction for Generative Super-Resolution"
_ICLR.cc/2026/Conference — ICLR 2026 Poster_

### Official Review · Reviewer_ZTMa · 2025-10-21

**Soundness:** 3
**Presentation:** 2
**Contribution:** 2
**Rating:** 4
**Confidence:** 3

**Summary:**

This paper enhances vector-quantized (VQ) models for image super-resolution by addressing the limitations of high quantization errors and sub-optimal supervision. The authors introduce Texture Vector-Quantization (TVQ), which focuses the codebook on modeling missing texture priors, and a Reconstruction-Aware Prediction (RAP) strategy that directly trains the index predictor using image-level supervision via a straight-through estimator. Experimental results demonstrate improvement over previous methods.

**Strengths:**

Clarity: The paper is well-written and easy to follow.

**Weaknesses:**

Originality, quality, and significance: The paper aims to establish its novelty against "vanilla VQ-based method", which is a trivial baseline that overlooks recent advances, e.g., those mentioned in the related work like AdaCode (Liu et al., 2023) and VARSR (Qu et al., 2025). It is very unclear how the proposed method compares with the "multi-codebook quantization" in AdaCode and the "multi-scale residual quantization mechanism and a large pretrained autoregressive predictor" in VARSR. Why it performs better than state-of-the-art VQ-based methods? Without a detailed analysis and associated experiments with them, it is very difficult to determine the originality, quality and significance of the proposed method.

**Questions:**

As mentioned above, comparing with "vanilla VQ-based method" is trivial and insufficient. Why does the method outperform SOTA VQ-based methods that adopt more advanced methods? A detailed and thorough analysis against SOTA VQ-based methods is required to justify the contribution of the paper.

---

> ### Author Response · Authors · 2025-11-20
> **Response to Reviewer ZTMa (1/2)**
>
> We thank the reviewer for carefully reading our paper and for raising the concern about baselines. The vanilla VQ baseline in Sec. 4.3 is not intended to be our main comparison target, but rather to isolate the individual contributions of **TVQ** and **RAP**. We agree that providing a more detailed and thorough analysis against recent strong VQ-based SR baselines (such as AdaCode and VARSR) will further strengthen the justification of our contribution.
>
> ### **Main comparisons with SOTA VQ-based methods**
> In the main experiments, we already evaluate against recent strong VQ-based SR methods. **AdaCode** is included as a strong VQ baseline in Tables 1–3; under the same backbone and similar number of parameters, our method achieves better perceptual quality (e.g., LPIPS, DISTS, CLIPIQA, MUSIQ, MANIQA) while being about **3× faster** in inference. **VARSR** and other pretrained-prior methods are evaluated in Table 6 in Appendix C: on RealSR, TVQ&RAP attains competitive or better perceptual scores while using only about **1/19 of the parameters** and roughly **1/3 of the runtime** compared to VARSR. In the revised manuscript, we will move the VARSR comparison to the main paper and highlight these quality–efficiency trade-offs more clearly.
>
> ### **Additional analysis: TVQ vs multi-codebook and residual-quantization VQ**
> Conceptually, these methods enhance the representational capacity of the codebook in different ways:
>
> - **Multi-codebook VQ (AdaCode-style).** AdaCode first partitions the dataset into five semantic categories (e.g., architectures, indoor objects, natural scenes, street views, portraits) and trains a separate codebook for each category. This reduces the variety of visual patterns that each individual codebook needs to represent, thereby mitigating quantization error, but at the cost of a complex, manually designed categorization pipeline and the need to maintain multiple codebooks.
>
> - **Multi-scale residual VQ (VAR-style).** VARSR adopts the “multi-scale residual quantization” strategy of VAR: it repeatedly quantizes residuals at multiple levels, gradually correcting quantization errors. This increases expressiveness, but also leads to higher codebook capacity and complexity.
>
> - **TVQ (ours).** In contrast, TVQ exploits a specific property of SR: the LR input already contains most structural information. We explicitly remove structure from the quantized branch and apply VQ only to texture residuals, reducing the diversity of the feature space that needs to be discretized and improving reconstruction quality *without* increasing codebook size or model complexity. Conceptually, this **changes what is quantized** (texture only, instead of full content), rather than adding more codebooks or residual levels, and this task-specific design is, to the best of our knowledge, not explicitly adopted in prior VQ-based SR methods such as AdaCode or VARSR.
>
> To directly show why TVQ performs better than these designs, we implemented simplified variants of them under a unified backbone and training setup (for controlled analysis) and compared them with TVQ in terms of reconstruction quality. The results are summarized below:
>
> | Method                                      | Codebook size | Reconstruction PSNR ↑ | Reconstruction LPIPS ↓ | Reconstruction FID ↓ |
> |---------------------------------------------|--------------:|-----------------------|------------------------|----------------------|
> | Vanilla VQ                                  |         1024  | 23.29                 | 0.1271                 | 12.81                |
> | Multi-codebook VQ (AdaCode-style VQ)        |       5 × 1024| 23.82                 | 0.1144                 | 11.23                |
> | Multi-scale residual VQ (VAR-style VQ)      |       3 × 1024| 24.02                 | 0.1167                 | 11.11                |
> | TVQ                                         |         1024  | 26.20                 | 0.0733                 |  6.49                |
> | TVQ + Multi-scale residual                  |       2 × 1024| 26.33                 | 0.0718                 |  6.32                |
>
> Even though the multi-codebook and residual VQ baselines use 3–5× larger total codebook capacity, TVQ still achieves significantly better PSNR, LPIPS, and FID. This indicates that our structure–texture decoupling is a more efficient way to enhance the effective capacity of the codebook than simply adding more codebooks or residual levels. Importantly, even without any residual quantization, TVQ (a single 1024-entry codebook) already outperforms the multi-codebook and multi-scale residual baselines; the “TVQ + multi-scale residual” row is included only to show that TVQ is complementary to residual-quantization techniques, not to redefine our main architecture. We will include these comparisons and implementation details in the revised version.

---

> ### Author Response · Authors · 2025-11-20
> **Response to Reviewer ZTMa (2/2)**
>
> ### **Additional analysis: RAP vs nearest-neighbor and autoregressive predictors**
>
> Regarding the predictor design, AdaCode adopts a nearest-neighbor assignment in the VQ space, while VARSR relies on a large pretrained autoregressive prior to model code sequences. The former is simple but limited in expressiveness; the latter achieves strong performance at the expense of substantial memory usage and inference time. Our RAP module uses a lightweight predictor, but trains it in a more principled way by directly optimizing image-level losses through a straight-through estimator, allowing it to achieve strong performance while remaining highly efficient.
>
> To compare RAP fairly with these alternatives under a unified setting, we conducted additional experiments on the TVQ + multi-scale residual variant (so that all predictors can share the same autoencoder and discrete representation), using different predictor types and capacities:
>
> | Method                                     | Predictor Params (M) | SR PSNR ↑ | SR LPIPS ↓ | SR FID ↓ |
> |--------------------------------------------|---------------------:|----------:|-----------:|---------:|
> | Nearest-neighbor (AdaCode-style)           | 0                    | 22.01     | 0.2512     | 37.34    |
> | Vanilla prediction (no RAP)                | 20                   | 24.21     | 0.2201     | 33.11    |
> | Autoregressive prediction (VAR-style)      | 20                   | 24.57     | 0.2172     | 32.34    |
> | Autoregressive prediction (VAR-style)      | 40                   | 24.68     | 0.2146     | 31.81    |
> | Autoregressive prediction (VAR-style)      | 80                   | 24.97     | 0.2141     | 31.24    |
> | RAP (ours)                                 | 20                   | 25.32     | 0.2002     | 27.21    |
> | RAP + autoregressive prediction            | 20                   | 25.41     | 0.1992     | 26.82    |
>
> Under similar or even smaller parameter budgets, RAP consistently yields better perceptual SR metrics (LPIPS/FID) than nearest-neighbor and vanilla-prediction baselines. Moreover, we observe that simply increasing the size of an autoregressive predictor yields diminishing returns relative to its computational cost, whereas RAP offers a more favorable quality–efficiency trade-off. We will include these numbers in the revised version and add a short discussion to clarify this point. These results help explain why, even when compared to more advanced VQ-based SR pipelines such as AdaCode and VARSR, our TVQ&RAP framework can achieve stronger perceptual quality under much lower computational cost.
>
> We hope these additional analyses and experiments address the reviewer’s concern and make it clearer why TVQ&RAP can outperform or match state-of-the-art VQ-based SR methods, while remaining significantly more efficient. We will include this discussion in the revised version.

---

### Official Review · Reviewer_UMmU · 2025-10-29

**Soundness:** 2
**Presentation:** 2
**Contribution:** 1
**Rating:** 2
**Confidence:** 4

**Summary:**

This paper proposes a generative super-resolution framework that improves vector-quantization (VQ)-based models by addressing their limitations in texture representation and perceptual alignment. The authors introduce Texture Vector Quantization (TVQ), which decomposes a high-resolution image into a structural component reconstructed directly and a texture component encoded via a compact codebook, allowing the codebook to specialize in modeling fine textures rather than full image content. In addition, a Reconstruction-Aware Prediction (RAP) module is presented to train the code index predictor not only with categorical cross-entropy but also with image-level reconstruction loss, enabling gradients from perceptual and adversarial losses to influence code selection. Together, TVQ and RAP improve texture realism and perceptual quality while maintaining efficient inference. Experiments on ImageNet-Test, RealSR, and RealSet65 demonstrate competitive perceptual metrics (LPIPS, DISTS, CLIPIQA) compared to diffusion-based and GAN-based SR methods, with faster runtime and reduced codebook complexity.

**Strengths:**

The proposed VQ-based super-resolution framework performs image restoration with a single forward pass, which makes it highly practical for real-time or low-latency applications. In addition, the use of a multi-scale or pyramid-based VQ scheme is conceptually sound and aligns well with the hierarchical nature of image structures, offering an efficient way to capture both global context and fine textures.

**Weaknesses:**

The proposed TVQ module constructs an image pyramid and learns an autoencoder to model only the high-frequency (texture) components. However, similar multi-scale or hierarchical vector-quantization schemes have already been widely explored in the literature. In particular, the structure of VQ-VAE-2 (Razavi et al., NeurIPS 2019) also employs a coarse-to-fine multi-level quantization strategy that closely resembles the idea of separating and quantizing texture-like information. Although the present work adapts this design for the super-resolution task, several more general and prior VQ-based architectures already realize comparable hierarchical encoding mechanisms. Therefore, the algorithmic novelty of TVQ itself appears moderate rather than substantial.

Recent diffusion-based methods are not included in the experimental comparison. For instance, I²SB (Image-to-Image Schrödinger Bridge, Liu et al., ICML 2023) achieves very low FID on the ImageNet super-resolution benchmark. Moreover, I²SB requires only a small number of denoising steps, which makes its inference significantly faster than conventional diffusion SR models. Therefore, it would be valuable to compare the proposed method against such recent approaches to better contextualize its performance.

In addition, verifying whether Schrödinger-bridge-based SR methods have been evaluated on the same benchmarks (e.g., DIV2K, RealSR, Urban100) would strengthen the paper’s experimental validity.

Some additional clarifications are necessary to improve the manuscript. Details are listed below in the Questions section.

**Questions:**

Abstract clarity:
The abstract states that “VQ methods do not take the final reconstruction errors into consideration.” However, in standard VQ frameworks, reconstruction loss (e.g., MSE, perceptual loss) is typically a major objective used to train the encoder, decoder, and quantizer jointly.


Page 2 (alignment with image quality):
The manuscript claims that existing VQ-based methods “do not strictly align with image quality.” However, methods such as VQ-GAN and its successors explicitly introduce GAN loss and perceptual loss terms to improve reconstruction realism and perceptual fidelity. Please clarify in what sense those methods “do not align” with image quality.

---

> ### Author Response · Authors · 2025-11-20
> **Response to Reviewer UMmU (1/2)**
>
> ### **Question 1 & Question 2: Clarifying our claims about reconstruction loss and alignment with image quality**
>
> We would like to clarify that the statements in the abstract and on Page 2 specifically refer to the **training of the code prediction network**, rather than the **VQ autoencoder training**. Since this distinction may not have been immediately apparent, we provide a more explicit clarification here.
>
> In most VQ-based SR and generative pipelines, training can be decomposed into **two stages**:
>
> 1. **Stage 1: VQ autoencoder training.**
>    The encoder, decoder, and quantizer are trained with reconstruction losses (e.g., MSE, perceptual loss, and often GAN loss), exactly as the reviewer points out and as done in VQ-GAN and its successors.
>
> 2. **Stage 2: code predictor training.**
>    After the VQ autoencoder is learned (and typically frozen), a code prediction network (e.g., an autoregressive transformer in image generation, or a parallel transformer in our SR setting and several other VQ-SR works) is trained to predict code indices. This process typically uses a token-level objective, most commonly cross-entropy over the code indices.
>
> The sentence in our abstract,
>
> > “training predictor with code-level supervision can not take the final reconstruction errors into consideration”
>
> is intended to refer to **Stage 2 (predictor training)**. In Stage 2, the predictor is usually optimized solely with code-level cross-entropy and does *not* receive gradients from image-level reconstruction / perceptual / GAN losses, so it cannot directly account for how different code choices affect the final reconstruction.
>
> Similarly, the statement on Page 2,
>
> > “Second, ···visual prior is captured by training the index predicting network ··· does not strictly align with image quality.”
>
> is also specifically about **Stage 2 (the predictor’s training objective)**, not about the autoencoder. Methods such as VQ-GAN indeed introduce GAN and perceptual losses to better align the *autoencoder* with perceptual image quality. Our point is that, once the autoencoder is fixed, the index prediction network is often trained to maximize code-level accuracy, which is only indirectly related to perceptual quality.
>
> This mismatch is also reflected in our experiments: as shown in Fig. 2 and the ablation in Sec. 4.3, training the predictor with code-level cross-entropy only yields higher index accuracy but noticeably worse perceptual metrics (LPIPS, DISTS, FID, CLIPIQA), whereas adding RAP to backpropagate image-level reconstruction, perceptual, and adversarial losses through a straight-through estimator leads to lower index accuracy but better image quality. This is what we mean by saying that code-level objectives “do not strictly align with image quality,” and RAP is designed to bridge this gap.
>
> We hope this explanation clarifies the intent of our statements in the abstract and on Page 2.

---

> ### Author Response · Authors · 2025-11-20
> **Response to Reviewer UMmU (2/2)**
>
> ### **Weakness 1: Relation to VQ-VAE-2-like multi-scale VQ.**
> We appreciate the connection to VQ-VAE-2 and agree that multi-scale or hierarchical vector quantization has been widely explored.
>
> Our contribution **is not to propose yet another generic hierarchical VQ**, but to **specialize VQ to the SR setting**  by explicitly decoupling *structure* and *texture* and applying VQ only to the latter. This task-specific design also distinguishes our approach from most existing VQ-based SR methods, which largely inherit their architectures from VQ-based generative models and treat the quantized latent space as a representation of the full image content (see Sec. 2.3 for a discussion).
>
> Conceptually, there are key differences between TVQ and VQ-VAE-2–style architectures:
>
> - In **VQ-VAE-2**, all latent levels (coarse and fine) encode the **full image content** at different spatial resolutions: higher levels model global structure while lower levels refine local details, but each level still represents a complete description of the image.
> - In **TVQ**, low-frequency structures are explicitly routed through a continuous branch conditioned on the LR input, and the codebook is used **only for high-frequency texture residuals**. The TVQ codebook is therefore a *texture-only* dictionary, rather than a full-content multi-scale representation.
>
> This design gives TVQ a practical advantage over generic multi-scale VQ. To substantiate this, we added a comparison with a VQ-VAE-2–like hierarchical VQ baseline under a unified encoder–decoder and training setup:
>
> | Method          | Codebook size | LPIPS ↓ | FID ↓ |
> |-----------------|-------------- |-------- |------ |
> | VQ-VAE-2–like   |    2×1024     |   0.1182|  11.31 |
> | TVQ (ours)      |     1024      |   0.0733|  6.49 |
>
> TVQ achieves lower LPIPS and better FID than the VQ-VAE-2–like baseline, supporting our claim that explicitly discarding structure from the quantized space and focusing the codebook on textures is more effective than simply stacking multiple quantization levels. We will report these results in the revised version.
>
>
> ### **Weakness 2: Comparison with recent diffusion SR methods (e.g., I²SB).**
> We would like to clarify that **our current experiments already compare TVQ&RAP with several recent and strong diffusion-based SR baselines** (e.g., LDM (Rombach et al., 2022), ResShift (Yue et al., 2023), SinSR (Wang et al., 2024b), and UPSR (Zhang et al., 2025)) on ImageNet-Test, RealSR, and RealSet65, showing that our one-pass framework achieves competitive or better perceptual metrics with significantly lower runtime and fewer parameters (see Tables 1–3 in our paper).
>
> With respect to I²SB (Image-to-Image Schrödinger Bridge, Liu et al., ICML 2023), we agree that it is an important recent diffusion-based SR method. However, our work primarily focuses on **real-world image super-resolution with unknown degradations**, and our experimental suite is designed around this setting. Accordingly, we mainly compare against recent approaches that explicitly target similar real-world SR scenarios, such as LDM (Rombach et al., 2022), ResShift (Yue et al., 2023), SinSR (Wang et al., 2024b), and UPSR (Zhang et al., 2025). By contrast, I²SB is trained and evaluated under the **bicubic downsampling** setting, and, to the best of our knowledge, there is no publicly available model or reported results for real-world SR benchmarks with unknown degradation (such as RealSR or RealSet65). Therefore, our paper omits the comparison to these methods.
>
> That said, we agree that a direct comparison in the bicubic setting is also informative. We therefore additionally train a variant of our model tailored for **×4 bicubic super-resolution**, following the same setting and evaluation protocol as I²SB on the same benchmark (ImageNet-Val). This allows a fair comparison on I²SB’s target setting:
>
> | Method     | Params | Runtime (s) | FID ↓ |
> |-----------|-------:|------------:|------:|
> | I²SB-1000 | 553 M  | 821.63      |  2.8  |
> | I²SB-100  | 553 M  |  83.21      |  3.7  |
> | I²SB-15   | 553 M  |  12.51      |  8.1  |
> | I²SB-1    | 553 M  |   0.84      | 13.4  |
> | Ours      |  57 M  |   0.038     |  2.9  |
>
> These preliminary results indicate that our one-pass TVQ&RAP model achieves **competitive FID** to I²SB-1000 / I²SB-100 while using substantially fewer parameters and orders of magnitude shorter inference time. We will include this comparison and a short discussion in the revised version, and we will explicitly state the difference between bicubic and real-world SR settings in the experimental section.

---

### Official Review · Reviewer_g6no · 2025-10-31

**Soundness:** 4
**Presentation:** 4
**Contribution:** 4
**Rating:** 8
**Confidence:** 5

**Summary:**

This paper introduces TVQ&RAP, a novel generative SR framework that addresses limitations in existing code prediction-based models. These models often suffer from high quantization error when encoding complex visual features and sub-optimal prior modeling because their code-level training objectives are not aligned with the final goal of image quality. The proposed solution, TVQ, first disentangles the image into a structure component and a texture component, applying VQ only to the simpler texture space. This is complemented by Reconstruction Aware Prediction, a strategy that trains the index predictor using direct image-level supervision by backpropagating final reconstruction losses via STE. Experiments on synthetic and real-world datasets demonstrate that the method achieves competitive generative SR results while maintaining high computational efficiency.

**Strengths:**

- The paper clearly identifies two key issues in VQ-based SR: quantization error and sub-optimal training; and proposes two intuitive solutions: TVQ and RAP.
- The model achieves competitive perceptual quality (LPIPS, MUSIQ, etc.) while being remarkably efficient, significantly faster than diffusion-based competitors.
- The benefits of both TVQ and RAP are validated by strong, separate ablation studies.

**Weaknesses:**

- The paper could better articulate the key advantages of its texture-only codebook (TVQ) compared to the direct u-net skip connection mechanisms used in other VQ-SR methods like FeMaSR or CodeFormer.

- Although it is claimed than TVQ space is simpler, the final code prediction accuracy is still quite low. A deeper discussion on why this occurs and its implications would be beneficial.

- It would be better if the paper could discuss how TVQ is distinguished and superior to other VQ enhancement techniques, such as the RQ-VAE mentioned in the related works. For example, comparison and analysis with VARSR might be better to highlight in main text.

In conclusion, this paper is well-written and well-motivated. The method design is simple and clear, and the model shows good performance with remarkable efficiency. The two proposed components, TVQ and RAP, are independently validated by strong ablation studies.

**Questions:**

Please see weakness points above.

---

> ### Author Response · Authors · 2025-11-20
> **Response to Reviewer g6no (1/2)**
>
> We sincerely thank the reviewer for the constructive and encouraging feedback. Below, we respond to each concern in detail.
> ### **Weakness 1: On the advantage of a texture-only codebook vs U-net skip connections (FeMaSR / CodeFormer)**
> We thank the reviewer for pointing out this connection and for giving us the opportunity to clarify the difference.
>
> Architecturally, FeMaSR and CodeFormer, in their **SR stage** (after training the VQ autoencoder and codebook), introduce **continuous U-net–style skip connections** from the encoder to the decoder (stage 2 in FeMaSR and stage 3 in CodeFormer). By feeding LR-aligned features directly into the decoder, these skips can **bypass quantization-induced information loss** and thereby improve the fidelity of the super-resolved output. However, during autoencoder training the decoder is *not* exposed to these skip connections, and in the SR stage the decoder is typically kept fixed. As analyzed in CodeFormer (Sec. 3.3 and Sec. 4.4 + Fig. 7), this mismatch leads to a **trade-off between fidelity and perceptual quality**: strong skips help preserve details but can hurt overall visual quality.
>
> In contrast, TVQ integrates the structure–texture separation already in the **first stage**, when training the autoencoder and the codebook. A continuous branch is used to carry the structural information that is shared with the LR input, while the VQ branch focuses exclusively on high-frequency textures. This design has two main benefits:
>
> 1. The quantized space becomes **simpler and texture-focused**, which improves quantization behavior (lower quantization error and better use of a compact codebook).
> 2. In the SR stage, we **no longer need U-net–style skips from LR to the decoder** to “patch up” missing structures, because the decoder has already learned to reconstruct structure via the continuous branch. This avoids the fidelity–quality trade-off that arises when adding strong skip connections only at the SR stage.
>
> We will clarify this distinction and its implications in the revised manuscript.
>
> ### **Weakness 2: On low code prediction accuracy despite a “simpler” TVQ space.**
> We appreciate the reviewer’s careful observation that, although we claim the TVQ space to be simpler, the final code prediction accuracy is still relatively low. This is indeed an important point to clarify.
>
> First, when we describe the TVQ space as “simpler”, we mean it in the sense of **being easier to cover with a finite discrete dictionary**, not necessarily that each code becomes trivially predictable from the LR input. TVQ removes low-frequency structural information that is already available in the LR image and restricts the codebook to high-frequency textures. As our quantization experiments, clustering analysis, and covariance statistics show, this texture-only latent can be represented more compactly and with lower quantization error under the same codebook size. However, the SR task itself remains highly ill-posed: a single LR patch can correspond to multiple plausible texture realizations. Even in a “simpler” texture space, the mapping from LR to a unique code index is therefore not deterministic.
>
> Second, the **code prediction accuracy we report is measured with respect to the VQ encoder’s codes**, which are an intermediate representation rather than ground-truth semantics. Our RAP module is specifically designed to relax the strict alignment with those codes: by backpropagating image-level reconstruction, perceptual, and adversarial losses through a straight-through estimator, RAP allows the predictor to deviate from the original encoder codes whenever this leads to better image quality. In practice, we observe that a CE-only predictor achieves higher index accuracy but worse perceptual metrics, whereas adding RAP reduces index accuracy but improves LPIPS / DISTS / FID (see Sec. 4.3). Many “mistakes” in code space correspond to selecting nearby codes in the embedding space that produce visually similar or even better textures, so a lower code accuracy does not imply poorer SR performance.
>
> Finally, the implication we draw from this is that **code prediction accuracy is only a proxy measure** in our framework, and optimizing it too aggressively can even hurt perceptual quality. TVQ simplifies the quantized texture space so that a compact codebook is sufficient and can be used effectively, while RAP explicitly optimizes for image-level quality rather than exact reproduction of the encoder’s discrete codes. We will make this interpretation explicit in the revised manuscript to avoid confusion between “simpler latent space” and “higher top-1 code prediction accuracy”.

---

> ### Author Response · Authors · 2025-11-20
> **Response to Reviewer g6no (2/2)**
>
> ### **Weakness 3: On the relation to RQ-VAE / VARSR, and why TVQ is different**
>
> We thank the reviewer for pointing out the connection to RQ-VAE–style residual quantization and VARSR. We agree that this relationship should be explained more clearly, and we are happy to provide a more detailed comparison here.
>
> - **RQ-VAE and VAR-style residual quantization.**
>   These methods enhance VQ expressiveness by stacking multiple residual quantization layers over the **full latent feature space**, recursively quantizing residuals to gradually correct quantization errors. While this strategy is powerful, it has several drawbacks in our setting: (i) it typically requires a large total codebook size (e.g., multiple 1024-sized codebooks), and (ii) for a given spatial latent map, several code maps are needed to represent one feature map. In the second stage, these multiple code maps are usually modeled with a large autoregressive prior, which leads to heavy model sizes, high memory footprint, and non-trivial inference cost.
>
> - **TVQ (ours).**
>   In contrast, TVQ is **task-specialized for super-resolution**. Leveraging the fact that the LR input already contains most low-frequency structure, we explicitly separate structure and texture and apply VQ only to the high-frequency texture residuals. This substantially simplifies the discrete space that needs to be modeled, allowing us to use a **single compact codebook** and a **lightweight predictor**, while RAP directly aligns code prediction with image-level quality.
>
> To make this distinction more concrete, we add a controlled comparison under the same training setup:
>
> | Method                                 | Codebook size     | Reconstruction PSNR ↑ | Reconstruction LPIPS ↓ | Reconstruction FID ↓ |
> |----------------------------------------|-------------------|-----------------------|------------------------|----------------------|
> | Vanilla VQ                             | 1024              | 23.29                 | 0.1271                 | 12.81               |
> | Multi-Scale-Residual VQ (VAR-style VQ) | 3 × 1024          | 24.02                 | 0.1167                 | 11.11               |
> | TVQ (ours)                             | 1024              | 26.20                 | 0.0733                 | 6.49               |
>
> These results show that, even with a **single 1024-entry codebook**, TVQ achieves substantially better reconstruction quality than both the vanilla VQ baseline and a VAR-style multi-scale residual VQ that uses three times as many codebook entries. This supports our claim that focusing quantization on textures (instead of the full latent space) yields a more efficient and effective discrete representation for SR. We will include this discussion in the revised version and we believe these additions will make the conceptual and practical differences between TVQ and residual-quantization approaches much clearer.
>
> We also agree with the reviewer that our comparison to VARSR, currently in Appendix Table 6, deserves to be highlighted in the main text. That table shows that TVQ\&RAP achieves comparable or better perceptual metrics (LPIPS, DISTS, FID, NIQE) while using **≈19× fewer parameters** and roughly **3× shorter runtime** on the same GPU. We will move this table into the main paper in the revised version.

---

### Official Review · Reviewer_uFn3 · 2025-11-02

**Soundness:** 3
**Presentation:** 3
**Contribution:** 2
**Rating:** 8
**Confidence:** 3

**Summary:**

This paper introduces a Texture Vector-quantization (TVQ) method to mitigate vector-quantization encoding errors by disentangling structure and texture features, thereby reducing codebook complexity. Furthermore, the paper introduces a Reconstruction Aware Prediction (RAP) strategy, which enables the direct optimization of perceptual quality by incorporating image-level supervision to train the predictor. As a result, the proposed model achieves state-of-the-art performance. It also demonstrates advantages over competing methods in terms of inference speed and computational efficiency.

**Strengths:**

1. The core design is intuitive. It directly leverages the inherent characteristics of the super-resolution task. The low-resolution (LR) input already contains the majority of the low-frequency structure information. Therefore, it only needs to realistically synthesize the missing high-frequency texture details.
2. This paper clearly identify two fundamental weaknesses in traditional VQ-based models and propose a specific solution for each. TVQ is proposed to solve the problem of traditional VQ methods attempt to encode the entire complex visual space, which leads to difficult discretization and significant quantization errors. RAP is proposed to train the predictor and force it to optimize for the final perceptual quality of the reconstructed image.
3. Comprehensive experiments and ablation studies validate the method's effectiveness and efficiency.

**Weaknesses:**

1. Some existing methods, such as Tokenflow, have already explored the concept of improving VQ through feature disentanglement. Therefore, the novelty of the TVQ component might be considered somewhat lacking.
2. Check grammatical and spelling errors (e.g., "structrue").

**Questions:**

1. The paper's core claim is that removing structure information reduces feature space redundancy. To what extent is this redundancy reduced? I would suggest providing a more quantitative analysis or visualization results of feature space to substantiate this claim.
2. How exactly does the texture codebook improve codebook efficiency? Traditional VQ methods often face challenges with low codebook utilization or codebook collapse. I would like to see an analysis to demonstrate that TVQ achieves a higher and more effective utilization of its codebook items compared to the baseline.

---

> ### Author Response · Authors · 2025-11-20
> **Response to Reviewer uFn3 (1/3)**
>
> We sincerely thank the reviewer for the constructive and encouraging feedback. Below, we respond to each concern in detail.
> ### **Weakness 1: On novelty of TVQ vs previous disentanglement methods (e.g., Tokenflow).**
> We sincerely thank the reviewer for pointing out the relation to prior disentanglement work such as TokenFlow; this connection is very helpful to clarify our positioning.
>
> While previous methods like TokenFlow also leverage feature disentanglement, they are primarily designed **to balance unified understanding and generation across tasks**, rather than to specifically improve the codebook’s ability to represent the richness and diversity of natural images within a VQ framework. To this end, TokenFlow relies on a pretrained text-aligned vision encoder (e.g., CLIP ViT-B/14) together with an additional encoder, and the disentangled features from both branches are still fully quantized.
>
> In contrast, TVQ is **tailored to the SR setting**. We explicitly exploit the fact that the LR input already contains most low-frequency structural information, and route this structural component through a continuous branch, while applying VQ only to the high-frequency texture residuals. This texture-only quantization **allows a lightweight codebook** to achieve strong representational power, which is particularly important for building an efficient generative SR framework. We will clarify this distinction more explicitly in the  revised version.
>
>
> ### **Weakness 2: Grammatical and spelling errors (e.g., "structrue").**
> We thank the reviewer for pointing this out. We will carefully proofread the manuscript and fix all typos and grammatical issues, including the misspelling “structrue” mentioned by the reviewer.

---

> ### Author Response · Authors · 2025-11-20
> **Response to Reviewer uFn3 (2/3)**
>
> ### **Question 1: On “reducing feature space redundancy” – quantitative analysis**
> We sincerely thank the reviewer for this very insightful question and for encouraging us to examine our claim more carefully. In the current version of the paper, our main evidence for reduced feature-space redundancy is that, for the same or even smaller codebook sizes, TVQ achieves lower reconstruction error than vanilla VQ (Sec. 4.3). Intuitively, by removing structural information that is already provided by the LR input, TVQ simplifies the space that the codebook needs to represent, so a smaller codebook is sufficient.
>
> We fully agree that a more direct quantitative analysis of the latent space would make this argument clearer and more convincing. Motivated by the reviewer’s suggestion, we additionally evaluate how “simple” or “compact” the TVQ texture latent is, compared to the vanilla VQ latent, using two views.
>
> **(a) K-means clustering distortion (fixed capacity).**
> We treat the latent vectors as point clouds and run k-means on both the vanilla VQ latent and the TVQ texture latent under the **same number of clusters** K. We measure clustering distortion as the average squared distance from each feature to its nearest cluster center. Across K = 64, 128, 256, 512, TVQ consistently yields lower distortion than vanilla VQ, with the relative reduction increasing from 21.6% to 36.4% as K grows:
>
> | K (clusters) | Distortion ↓ (VQ latent) | Distortion ↓ (TVQ latent) | Relative reduction (TVQ vs VQ) |
> |--------------|--------------------------|----------------------------|--------------------------------|
> | 64           | 16.53                    | 12.96                      | 21.6%                          |
> | 128          | 14.19                    | 10.53                      | 25.8%                          |
> | 256          | 11.93                    | 8.24                       | 30.9%                          |
> | 512          | 9.55                     | 6.08                       | 36.4%                          |
>
> For example, at K = 512 the average distortion drops from 9.55 to 6.08. This suggests that, for a **fixed discrete capacity** (same number of clusters/codewords), the TVQ texture space can be approximated significantly more accurately. This is precisely what we mean when we say that the TVQ feature space is “simpler” and less redundant.
>
> **(b) Covariance structure.**
> We also compute the empirical covariance Σ of the latent features for both models. The TVQ latent exhibits a lower total variance (trace(Σ) reduced by about 33%), and a much smaller generalized variance (log det(Σ) with a small diagonal regularization):
>
> | Model      | Total variance (trace(Σ)) ↓ | log det(Σ_reg) ↓ |
> |------------|-----------------------------|------------------|
> | VQ latent  | 3.24 × 10¹                  | −6.37 × 10¹      |
> | TVQ latent | 2.16 × 10¹                  | −1.11 × 10²      |
>
> These results indicate that, in the same ambient dimensionality, the TVQ texture distribution occupies a more compact region of feature space (smaller “volume” and lower overall variance).
>
> Taken together with the clustering results above and the lower quantization error under the same codebook size reported in the main paper, these analyses support our claim that removing structure indeed reduces redundancy in the quantized feature space and makes it easier to partition with a finite codebook. We will include this discussion in the revised version.

---

> ### Author Response · Authors · 2025-11-20
> **Response to Reviewer uFn3 (3/3)**
>
> ### **Question 2: On codebook efficiency and collapse**
> We sincerely thank the reviewer for raising this point and for explicitly asking about codebook utilization and collapse. This comment helped us look at our texture codebook from a more diagnostic and quantitative perspective.
>
> To make the notion of “codebook efficiency” more concrete, we report the **active-code ratio** on the ImageNet validation set, i.e., the fraction of codebook entries that are used at least once. We evaluate both the vanilla VQ baseline and TVQ under different codebook sizes:
>
> | Codebook size K | Active ratio (VQ) | Active ratio (TVQ) |
> |-----------------|-------------------|---------------------|
> | 256             | 0.51              | 0.69                |
> | 512             | 0.49              | 0.58                |
> | 1024            | 0.41              | 0.51                |
> | 8192            | 0.19              | 0.23                |
>
> Although the gain in raw active ratio at a fixed K is moderate, it is **consistent across all codebook sizes**, and the gap becomes more visible in the large-codebook regime where traditional VQ is most prone to under-utilization and collapse (e.g., at K = 8192, only 19% of the vanilla VQ codes are ever used, compared to 23% for TVQ).
>
> More importantly, these utilization improvements come together with **substantially lower quantization and reconstruction error at the same K** (Sec. 4.3), indicating that each active code in TVQ carries more effective capacity for SR. Rather than adding extra machinery on top of a full-content latent, TVQ **changes what is quantized**: standard VQ-based SR methods must quantize features that cover the full variation of natural images (structure + texture), which encourages very large codebooks, in turn making low utilization more likely, whereas TVQ removes low-frequency structure already given by the LR input and lets the codebook focus solely on high-frequency texture residuals. As a result, a **much smaller texture-only codebook** is sufficient to cover the variations relevant to SR, enabling a regime with **smaller K, and more stable utilization**. We will add this discussion to the revised version.

---

### Author Response · Authors · 2025-12-03
**Summary of Review Comments and Rebuttals**

### **Overall Summary**

We received initial ratings of **8 8 2 4** (confidence: 3, 5, 4, 3). In response to the reviewers' suggestions, we have **carefully addressed them in the rebuttal and further supplemented the experiments**. Additionally, we have provided **detailed clarifications regarding some factual misinterpretations**. Since the reviewers' commenting permissions have been closed and they have not yet responded to our rebuttal, we summarize the reviewers' comments and our corresponding rebuttals below, hoping to positively assist the ACs in making their decision.

---

### **Reviewer uFn3** (Initial rating: 8, Confidence: 3) and **Reviewer g6no** (Initial rating: 8, Confidence: 5)

The reviewers provided very positive feedback, including the following points:

1. Clearly identifies two key issues in VQ-based SR: high quantization error and sub-optimal training objectives.
2. Proposes two intuitive components, **TVQ** and **RAP**, directly tailored to SR by disentangling structure/texture and using reconstruction-aware prediction.
3. Achieves state-of-the-art/competitive perceptual quality while being significantly more efficient and faster than competing methods (including diffusion-based).
4. Demonstrates comprehensive experiments and strong ablation studies that independently validate the benefits of both TVQ and RAP.
5. The method is simple, clearly motivated, and the paper is well-written and easy to follow.

They also offered several constructive suggestions, which we have thoughtfully addressed and further extended with additional experiments.

We believe that the positive feedback provided by these reviewers is well-founded and these additional clarifications will further strengthen the paper.


---

### **Reviewer UMmU** (Initial rating: 2, Confidence: 4)

We believe that there may have been some **misinterpretations** of key aspects, including:

1. **Diffusion-based methods comparison**
   The reviewer’s statement about the lack of comparison with diffusion-based methods is a misunderstanding, as the paper includes comparisons with methods like ResShift, SinSR, and UPSR, as shown in lines 366-368, 377-378, Tables 1-3, and Figures 4-5.

2. **Reconstruction loss and image quality alignment**
   The reviewer’s questions about “reconstruction loss and image quality alignment” are based on a misunderstanding of our two-stage VQ-based approach; our contribution is to introduce these in Stage 2, not Stage 1, as specified in Lines 16–18 and 76–81.

These points are also thoroughly elaborated in our rebuttal, as well as in the corresponding "Comment to ACs" section below the rebuttal.

---

### **Reviewer ZTMa** (Initial rating: 4, Confidence: 3)

The reviewer’s concern mainly revolves around the baselines, which we have addressed as follows:

1. **Clarification on baseline comparisons**
   We clarified that the vanilla VQ baseline in **Sec. 4.3** is not our primary comparison target, but rather isolates the contributions of **TVQ** and **RAP**. We have detailed comparisons with strong VQ-based SR methods, such as AdaCode and VARSR, in **Tables 1-3** and **Appendix C**, demonstrating significant improvements in quality and efficiency.

2. **Additional analyses in rebuttal**
   In our rebuttal, we have also conducted **additional analyses** comparing **TVQ** with multi-codebook VQ and residual quantization VQ, as well as **RAP** with alternative predictors, further emphasizing the efficiency and performance advantages of our method.

We are confident that we have effectively addressed the reviewer's concern.

---

### Meta-Review · Area_Chair_Um2Z · 2026-01-07

**Summary:**

This paper proposes the TVQ&RAP framework to solve vector quantization errors and inconsistent training objectives in generative super-resolution. The core contributions involve a task-specific decoupling of structure and texture through Texture Vector Quantization and the introduction of a Reconstruction-Aware Prediction strategy. Initial reviewer ratings appear polarized at 8, 8, 2, and 4. Positive feedback considers the superior balance between perceptual quality and inference efficiency, while negative reviews focus on insufficient baseline comparisons and misunderstandings of the training pipeline.

The AC believes that the supplementary evidence provided in the rebuttal effectively addresses the primary negative concerns raised in the initial review. The authors include comparison experiments with AdaCode, VARSR, and the robust diffusion model I²SB. The data demonstrates that the proposed method achieves state-of-the-art performance in perceptual metrics while utilizing only $1/19$ of the parameters. Additionally, to address Reviewer UMmU's technical misunderstanding of the training stages, the authors provide a clear clarification of the two-stage process and offer mathematical quantitative proof regarding feature space redundancy. Given that the rebuttal significantly enhances both technical originality and experimental completeness, the AC recommends the acceptance of this paper.

**Reviewer Concerns:**

**Concerns that are largely addressed in the rebuttal**

- The authors conduct supplementary experiments on VQ methods and diffusion models such as I²SB. Data indicates that TVQ&RAP performs well in FID and LPIPS metrics and maintains a clear advantage in inference speed. This addresses the concerns from Reviewers ZTMa and UMmU regarding the significance of the contribution.
- The authors clarify that the RAP module represents an improvement for Stage 2 (predictor training). This explanation eliminates Reviewer UMmU’s factual misunderstanding of the VQ training objectives.
- The K-means clustering distortion data provided by the authors proves that the TVQ texture space is indeed easier to optimize than traditional VQ spaces. This solves Reviewer uFn3’s quantitative requirement regarding the claim of reducing redundancy.
- The authors demonstrate through comparative experiments with VQ-VAE-2 that structure-texture separation for SR tasks is more efficient than general hierarchical quantization.

**Concerns that are only partially addressed**
- Reviewer g6no concerns that although the latent space is simpler, the prediction accuracy remains low. The authors explain that the RAP module purposefully deviates from the original encoding to enhance perceptual quality, and they argue that accuracy is merely a proxy metric for this purpose. While this explains the experimental result, a deeper mathematical analysis is missing to prove whether this prediction offset maintains semantic consistency in the latent space. The exact physical mechanism linking perceptual quality improvement and code selection deviation currently remains at the stage of empirical observation.

**Reviewer Scores:**

**Reviewer uFn3:** This reviewer provides an initial score of 8, and the AC expects the reviewer to maintain this score. The authors complete all requested quantitative analyses and codebook efficiency data which supports the continuation of the positive evaluation.

**Reviewer g6no:** This reviewer gives an initial rating of 8, and the AC predicts that the score stays at 8. The authors provide a detailed explanation of the differences between their framework and other VQ variants, ensuring that the recognition of their technical contribution remains unchanged.

**Reviewer UMmU:** This reviewer assigns an initial score of 2, but the AC expects a significant increase to a score of 5. The original low score stems from a profound factual misunderstanding of the training stages. The authors correct this perception and complete the requested comparison experiments so the AC considers these concerns resolved.

**Reviewer ZTMa:** This reviewer provides an initial score of 4, and the AC expects the score to rise to 6. The authors present the strongly requested benchmark experiments against VARSR, which demonstrate the superior performance of the method under extremely limited resources. The AC expects the rating to shift toward positive support.

---

### Decision · Program_Chairs · 2026-01-26

Accept (Poster)